# SEAL: A Framework for Systematic Evaluation of Real-World Super-Resolution

**Wenlong Zhang[1,2], Xiaohui Li[2,3], Xiangyu Chen[2,4,5], Xiaoyun Zhang[3]**
**Yu Qiao[2,5], Xiao-Ming Wu[1]\*, Chao Dong[2,5]\***
[1]The HongKong Polytechnic University, [2]Shanghai AI Laboratory [3]Shanghai Jiao Tong University
[4]University of Macau [5]Shenzhen Institute of Advanced Technology, CAS
`wenlong.zhang@connect.polyu.hk, xiao-ming.wu@polyu.edu.hk, chao.dong@siat.ac.cn`

## Abstract

Real-world Super-Resolution (Real-SR) methods focus on dealing with diverse real-world images and have attracted increasing attention in recent years. The key idea is to use a complex and high-order degradation model to mimic real-world degradations. Although they have achieved impressive results in various scenarios, they are faced with the obstacle of evaluation. Currently, these methods are only assessed by their average performance on a small set of degradation cases randomly selected from a large space, which fails to provide a comprehensive understanding of their overall performance and often yields inconsistent and potentially misleading results. To overcome the limitation in evaluation, we propose SEAL, a framework for systematic evaluation of real-SR. In particular, we cluster the extensive degradation space to create a set of representative degradation cases, which serves as a comprehensive test set. Next, we propose a coarse-to-fine evaluation protocol to measure the distributed and relative performance of real-SR methods on the test set. The protocol incorporates two new metrics: acceptance rate ($AR$) and relative performance ratio ($RPR$), derived from acceptance and excellence lines. Under SEAL, we benchmark existing real-SR methods, obtain new observations and insights into their performance, and develop a new strong baseline. We consider SEAL as the first step towards creating a comprehensive real-SR evaluation platform, which can promote the development of real-SR. The source code is available at `https://github.com/XPixelGroup/SEAL`

## 1 Introduction

Image super-resolution (SR) aims to reconstruct high-resolution (HR) images from their low-resolution (LR) counterparts. Recent years have witnessed great success in classical SR settings (i.e., bicubic downsampling) with deep learning techniques (Dong et al., 2014; Zhang et al., 2018b;c; Ledig et al., 2017; Wang et al., 2018). To further approach real-world applications, a series of "blind" SR methods have been proposed to deal with complex and unknown degradation kernels (Zhang et al., 2018a; Gu et al., 2019; Wang et al., 2021a; Luo et al., 2020). Among them, real-SR methods, such as BSRGAN (Zhang et al., 2021) and RealESRGAN (Wang et al., 2021b), have attracted increasing attention due to their impressive results in various real-world scenarios.

Different from classical SR that only adopts a simple downsampling kernel, real-SR methods propose complex degradation models (e.g., a sequence of blurring, resizing and compression operations) that can represent a much larger degradation space, covering a wide range of real-world cases. However, they also face a dilemma in evaluation: *As the degradation space is vast, how to evaluate their overall performance?* Directly testing on all degradations is obviously infeasible, as there are numerous degradation combinations in the vast degradation space.

To evaluate the performance of real-SR methods, previous works directly calculate the average performance on a randomly-sampled small-sized test set based on an IQA metric (e.g., PSNR). However, we find that this evaluation protocol is fatally flawed. Due to the vastness of the degradation

---

*Corresponding author

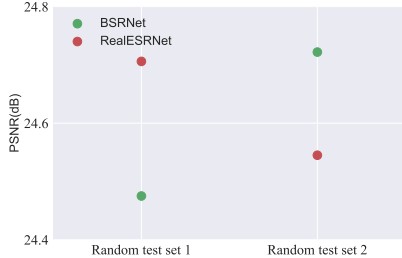

(a) Results with the conventional evaluation approach.

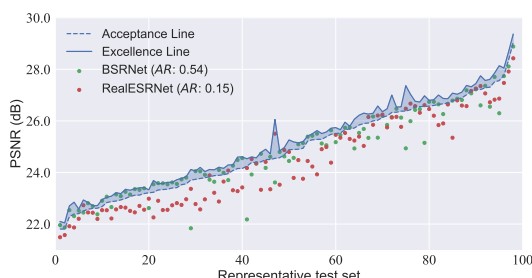

(b) Results with our framework SEAL.

Figure 1: (a) We compare the average performance of BSRNet and RealESRNet on two real test sets generated by common practice. There is a significant variance in their performance: the differences between their average PSNR on the two test sets are -0.23dB and 0.18dB respectively, leading to contradictory conclusions. (b) BSRNet and RealESRNet assessed under our SEAL framework in a distributed manner with 100 representative test sets. It shows the former outperforms the latter in 60% cases, providing a comprehensive overview of their performance.

space, a small test set that is selected randomly cannot reliably represent the degradation space and may cause significant bias and randomness in the evaluation results, as illustrated in Fig. 1. In addition, the current evaluation strategy is not enough for assessing real-SR methods, as they typically average quantitative results across all testing samples, which may also lead to misleading comparison results. For example, one method may outperform another on 60% of the degradation types, but it may not achieve a higher mean PSNR value for the entire test set (Sec. 5.3). The average score cannot adequately represent the overall performance and distribution. Furthermore, if the goal is to improve the average score, we could focus solely on enhancing the performance of simple cases (e.g., small noise or blur), which, however, would adversely affect difficult ones (e.g., complex degradation combinations). This would contradict our main objective. Instead, once we have achieved satisfactory outcomes in easy cases, we should divert our focus towards challenging ones to enhance the overall performance. The aforementioned points indicate the need for a new framework that can comprehensively evaluate the performance of real-SR methods.

In this work, we establish a **s**ystematic **e**valuation framework for re**al**-SR, namely SEAL, which assesses relative, distributed, and overall performance rather than relying solely on absolute, average, and misleading evaluation strategy commonly used in current evaluation methods. Our first step is to use a clustering approach to partition the expansive degradation space to identify representative degradation cases, which form a comprehensive test set. In the second step, we propose an evaluation protocol that incorporates two new relative evaluation metrics, namely Acceptance Ratio ($AR$) and Relative Performance Ratio ($RPR$), with the introduction of acceptance and excellence lines. The $AR$ metric indicates the percentage by which the real-SR method surpasses the acceptance line, which is a minimum quality benchmark required for the method to be considered satisfactory. The $RPR$ metric measures the improvement of the real-SR method relative to the distance between the acceptance and excellence lines. The integration of these metrics intends to provide a more thorough and detailed evaluation of real-SR methods.

With SEAL, it becomes possible to conduct a comprehensive evaluation of the overall performance of real-SR methods, as illustrated in Fig. 1. The significance of our work can be summarized as:

- Our relative, distributed evaluation approach serves as a complement to existing evaluation methods that solely rely on absolute, average performance, addressing their limitations and providing a valuable alternative perspective for evaluation.

- By employing SEAL, we benchmark existing real-SR models, leading to the discovery of new observations and valuable insights, which further enables us to develop a new strong real-SR model.

- The components of our SEAL framework are flexibly customizable, including the clustering algorithm, acceptance/excellence lines, and evaluation protocol. It can facilitate the development of appropriate test sets and comparative evaluation metrics for real-SR.

## 2 RELATED WORK

**Image super-resolution.** Since Dong *et al.* (Dong et al., 2014) first introduced Convolutional Neural Networks (CNNs) to the Super-Resolution (SR) task, there have been significant advancements in the field. A variety of techniques have been developed, including residual networks (Kim et al., 2016), dense connections (Zhang et al., 2018c), channel attention (Zhang et al., 2018b), residual-in-residual dense blocks (Wang et al., 2018), and transformer structure (Liang et al., 2021a; Chen et al., 2023b). To reconstruct realistic textures, Generative Adversarial Networks (GANs) (Ledig et al., 2017; Wang et al., 2018; Zhang et al., 2019; Wenlong et al., 2021) are introduced to SR approaches for generating visually pleasing results. Although these methods have made significant progress, they often rely on a simple degradation model (i.e., bicubic downsampling), which may not adequately recover the low-quality images in real-world scenarios.

**Blind super-resolution.** Several works have been made to improve the generalization of SR networks in real-world scenarios. These works employ multiple degradation factors (e.g., Gaussian blur, noise, and JPEG compression) to formulate a blind degradation model. SRMD (Zhang et al., 2018a) employs a single SR network to learn multiple degradations. Kernel estimation-based methods (Gu et al., 2019; Luo et al., 2020; Bell-Kligler et al., 2019; Wang et al., 2021a) introduce a kernel estimation network to guide the SR network for the application of the low-quality image with different kernels. To cover the diverse degradations of real images, BSRGAN (Zhang et al., 2021) proposes a practical degradation model that includes multiple degradations with a shuffled strategy. RealESRGAN (Wang et al., 2021b) introduces a high-order strategy to construct a large degradation model. These works demonstrate the potential of blind SR in real-world applications.

**Model evaluation for super-resolution.** For non-blind SR model evaluation, a relatively standard process employs the fixed bicubic down-sampling on the benchmark test datasets to generate low-quality images. However, it is typically implemented using a predefined approach (e.g., uniform sampling) for blind SR, such as the general Gaussian blur kernels (Zhang et al., 2018a; Liang et al., 2021c), Gaussian8 kernels (Gu et al., 2019), and five spatially variant kernel types (Liang et al., 2021b). For real-SR, existing methods often add random degradations to DIV2K_val (includes 100 Ground-Truth images) to construct the real test set, such as DIV2K4D in BSRGAN (Zhang et al., 2021) and DIV2K_val with three Levels in DASR (Liang et al., 2022). However, these methods use small test sets with average performance, making it difficult to evaluate overall performance across different degradation combinations in real-world scenarios.

## 3 DEGRADATION SPACE MODELING

### 3.1 GENERATING THE DEGRADATION SPACE

In real-SR, the degradation process (Wang et al., 2021b) can be simulated by

$$I^{\text{LR}} = (d_s \circ \cdots \circ d_2 \circ d_1)(I^{\text{HR}}), \tag{1}$$

where $s$ is the number of degradations applied on a high-resolution image $I^{\text{HR}}$, and $d_i$ ($1 \leq i \leq s$) represents a randomly selected degradation. Assume there are only $s$ degradation types (e.g., blur, resize, noise, and compression), and each type contains only $k$ discrete degradation levels. The total degradation should be $A_s^s \times k^s$. With $s = 10$ and $k = 10$, it will generate a degradation space of magnitude $(A_{10}^{10}) * 10^{10}$, which is already an astronomical figure. Clearly, randomly sampling a limited number of degradations (e.g., 100 in existing works (Zhang et al., 2021)) from such a huge space cannot adequately represent the entire space, which will inevitably result in inconsistent and potentially misleading outcomes, as illustrated in Fig. 1.

### 3.2 REPRESENTING THE DEGRADATION SPACE

To represent the degradation space $\mathbb{D}$, a straightforward way is to divide the space by degradation parameters, which may seem reasonable at first glance. However, we observe that different combinations of degradation types may have similar visual quality and restoration difficulty. As shown in Fig. 13 of the Appendix, the images undergone different degradations have similar appearances. This suggests that it might be more reasonable to distinguish the degraded images with their low-level features instead of degradation parameters.

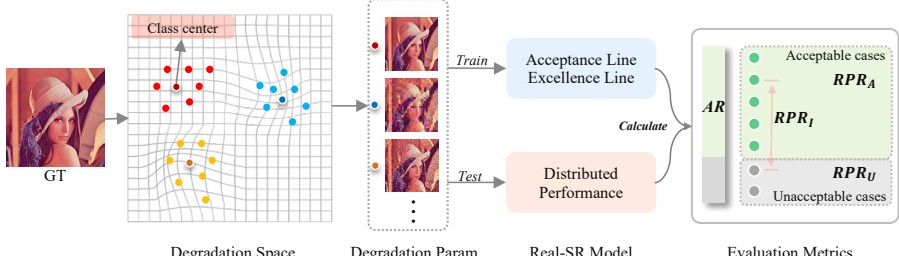

Figure 2: **Our proposed evaluation framework** consists of a clustering-based approach for degradation space modeling (Sec. 3) and a set of metrics based on representative degradation cases (Sec. 4). We divide the degradation space into $K$ clusters and use the degradation parameters of the class centers to create $K$ training datasets to train $K$ *tiny/large* SR models as the acceptance/excellence line. The distributed performance (Eq. 4) of the real-SR model across the $K$ test datasets will be compared with the acceptance and excellence lines and evaluated by a set of metrics.

Therefore, we propose to find prototypical degradation cases to represent the vast degradation space. As shown in Fig. 2, a plausible solution is to cluster the degradation space by grouping the degraded images into $K$ groups and choose the $K$ group centers as the representative cases:

$$\mathcal{D} = \{c_1, c_2, \cdots, c_K\}, \tag{2}$$

where $c_i(1 \leq i \leq K)$ is the center of the $i$-th group. Note that the images can be represented by their features (e.g., image histograms) and clustered by a conventional clustering algorithm such as spectral clustering.

### 3.3 EVALUATING REAL-SR MODELS USING THE REPRESENTATIVE DEGRADATION CASES

We then use the degradation parameters of the cluster centers $c_i$ to construct a test set for systematic evaluation, denoted as the SE test set:

$$\mathcal{D}^{\text{test}} = \{\mathcal{D}_{c_1}, \mathcal{D}_{c_2}, \cdots, \mathcal{D}_{c_K}\}, \tag{3}$$

where $\mathcal{D}_{c_i}(1 \leq i \leq K)$ is a set of low-quality images obtained by using the degradation parameters of $c_i$ on a set of clean images (e.g., DIV2K dataset). $\mathcal{D}_{c_i}$ can be used to evaluate the distributional performance of a real-SR model on a representative degradation case. $\mathcal{D}^{\text{test}}$ can be used to provide a full picture of the performance on all representative degradation cases.

## 4 EVALUATION METRICS

To provide a comprehensive and systematic overview of the performance of a real-SR model on $\mathcal{D}^{\text{test}}$, we develop a set of evaluation metrics to assess its effectiveness in a quantitative manner.

**Distributed Absolute Performance.** The most straightforward way to evaluate a real-SR model is to compute its distributed performance on $\mathcal{D}^{\text{test}}$:

$$\mathcal{Q}^{\text{d}} = \{Q_1^{\text{d}}, Q_2^{\text{d}}, \cdots, Q_K^{\text{d}}\}, \quad Q_{\text{ave}}^{\text{d}} = \frac{1}{K}\sum_i Q_i^{\text{d}}, \tag{4}$$

where $Q_i^{\text{d}}$ represents the average absolute performance of a real-SR model on the $i$-th representative test set $\mathcal{D}_i$, and $Q_{\text{ave}}^{\text{d}}$ denotes the average absolute performance on $\mathcal{D}^{\text{test}}$.

**Distributed Relative Performance.** To comparatively evaluate a real-SR model and pinpoint the representative cases where its performance is deemed inadequate, we specify an acceptance line and an excellence line. The acceptance line is designated by a small network (e.g., FSRCNN (Dong et al., 2016)), while the excellence line is provided by a large network (e.g., SRResNet (Ledig et al., 2017)). If the real-SR model is unable to surpass the small network on a representative case, it is considered failed on that case. We use $\mathcal{Q}^{\text{ac}}$ and $\mathcal{Q}^{\text{ex}}$ to represent the acceptance line and the excellence line:

$$\mathcal{Q}^{\text{ac}} = \{Q_1^{\text{ac}}, Q_2^{\text{ac}}, \cdots, Q_K^{\text{ac}}\}, \quad \mathcal{Q}^{\text{ex}} = \{Q_1^{\text{ex}}, Q_2^{\text{ex}}, \cdots, Q_K^{\text{ex}}\}, \tag{5}$$

where $Q_i^{\text{ac}}$ and $Q_i^{\text{ex}}$ are the performance of the small and large networks trained with the degradation parameters of $c_i$ in a non-blind manner.

**Acceptance Rate** (AR) measures the percentage of acceptable cases among all $K$ representative degradation cases for a real-SR model. An acceptable case is one in which the performance of a real-SR model surpasses the acceptance line. $AR$ is defined as

$$AR = \frac{1}{K} \sum_i \mathbb{I}(Q_i^{\text{d}} > Q_i^{\text{ac}}), \tag{6}$$

where $\mathbb{I}$ represents the indicator function. $AR$ can reflect the overall generalization ability of a real-SR model.

**Relative Performance Ratio** ($RPR$) is devised to compare the performance of real-SR models at the same scale w.r.t. the acceptance and excellence lines. It is defined as

$$RPR_i = \sigma\left(\frac{Q_i^{\text{d}} - Q_i^{\text{ac}}}{Q_i^{\text{ex}} - Q_i^{\text{ac}}}\right), \quad \text{and} \quad \mathcal{R} = \{RPR_1, RPR_2, \cdots, RPR_K\}, \tag{7}$$

where $\sigma$ denotes the sigmoid function, which is used to map the value to (0, 1). Note that $RPR_i > \sigma(0) = 0.5$ indicates that the real-SR model is better than the acceptance line on the $i$-th degradation case, and $RPR_i > \sigma(1) = 0.73$ means it is better than the excellence line.

a) **Interquartile range of** $RPR$ ($RPR_I$) is used to access the level of variance in the performances of a real-SR model on $\mathcal{D}^{\text{test}}$. It is defined as:

$$RPR_I = \mathcal{R}_{W_3} - \mathcal{R}_{W_1}, \tag{8}$$

where $\mathcal{R}_{W_3}$ and $\mathcal{R}_{W_1}$ denote the $75^{th}$ and $25^{th}$ percentiles (Wan et al., 2014) of the $RPR$ scores, respectively. Low $RPR_I$ means the real-SR model demonstrates a similar relative improvement in most degradation cases.

b) **Average** $RPR$ **on acceptable cases** ($RPR_A$) computes the mean of $RPR$ scores on acceptable cases:

$$RPR_A = \frac{1}{|\mathcal{R}_A|} \sum_{\mathcal{R}_i \in \mathcal{R}_A} \mathcal{R}_i, \quad \mathcal{R}_A = \{\mathcal{R}_i \in \mathcal{R} | \mathcal{R}_i \geq 0.5\}. \tag{9}$$

Note that $RPR_A \in (0.5, 1)$, and $RPR_A > 0.73$ means the average performance of a real-SR model on acceptable cases exceeds the excellence line.

c) **Average** $RPR$ **on unacceptable cases** ($RPR_U$) computes the mean of $RPR$ scores on unacceptable cases:

$$RPR_U = \frac{1}{|\mathcal{R}_U|} \sum_{\mathcal{R}_i \in \mathcal{R}_U} \mathcal{R}_i, \quad \mathcal{R}_U = \{\mathcal{R}_i \in \mathcal{R} | \mathcal{R}_i < 0.5\}. \tag{10}$$

Note that $RPR_U \in (0, 0.5)$, and $RPR_A$ near 0.5 means the average performance of a real-SR model on unacceptable cases is close to the acceptance line.

**Coarse-to-fine Evaluation Protocol.** Based on the proposed metrics, we develop a coarse-to-fine evaluation protocol to rank different real-SR models. As illustrated in Fig. 3, the models are compared by the proposed metrics sequentially by order of priority. $AR$ represents a coarse-grained comparison, while $RPR$ provides a fine-grained comparison. If their performances are too close to the current metric, the next metric is used to rank them.

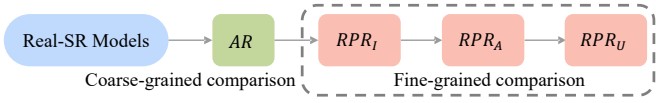

Figure 3: **A coarse-to-fine evaluation protocol** to rank real-SR models with the proposed metrics.

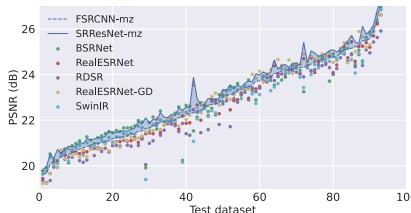

Figure 4: **Visualization of distributed performance** in PSNR for MSE-based real-SR methods on Set14-SE.

Table 1: **Benchmark results and ranking of MSE-based real-SR methods** in PSNR by our proposed SEAL. The subscript denotes the rank order. $\times$ means the model fails in a majority of degradation cases.

| Models | PSNR | $AR\uparrow$ | $RPR_I\downarrow$ | $RPR_A\uparrow$ | $RPR_U\uparrow$ | Rank |
|---|---|---|---|---|---|---|
| SRResNet | 20.95 | $0.00_{(\times)}$ | 0.02 | 0.00 | 0.03 | $\times$ |
| DASR | 21.08 | $0.00_{(\times)}$ | 0.01 | 0.00 | 0.02 | $\times$ |
| BSRNet | $22.77_{(2)}$ | $0.59_{(1)}$ | $0.42_{(4)}$ | $0.72_{(2)}$ | $0.27_{(4)}$ | 1 |
| RealESRNet | $22.67_{(3)}$ | $0.27_{(4)}$ | $0.28_{(2)}$ | $0.63_{(3)}$ | $0.28_{(3)}$ | 4 |
| RDSR | $22.44_{(5)}$ | $0.08_{(\times)}$ | 0.23 | 0.63 | 0.21 | $\times$ |
| RealESRNet-GD | $22.82_{(1)}$ | $0.43_{(2)}$ | $0.37_{(3)}$ | $0.74_{(1)}$ | $0.33_{(1)}$ | 2 |
| SwinIR | $22.61_{(4)}$ | $0.41_{(3)}$ | $0.24_{(1)}$ | $0.58_{(4)}$ | $0.29_{(2)}$ | 3 |

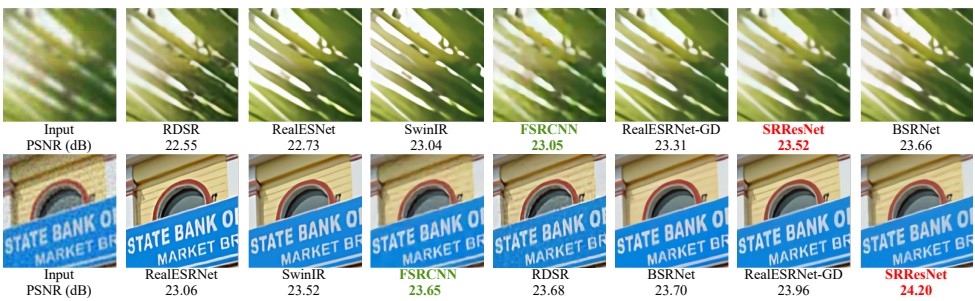

Figure 5: **Visual results of MSE-based real-SR methods** with the acceptance line FSRCNN and excellence line SRResNet. It is best viewed in color.

# 5 EXPERIMENTS

## 5.1 IMPLEMENTATION

**Constructing the test set for systematic evaluation.** We utilize two widely-used degradation models, BSRGAN (Zhang et al., 2021) and RealESRGAN (Wang et al., 2021b), which are designed to simulate the real-world image space. By combining the two degradation models with equal probability [0.5, 0.5], we generate a dataset of $1 \times 10^4$ low-quality images from the ground-truth (GT) image, *lenna*, from the Set14 dataset (Zeyde et al., 2010). To categorize the degraded images, we employ spectral clustering due to its effectiveness in identifying clusters of arbitrary shape, making it a flexible choice for our purposes. Specifically, we first use the histogram feature (Tang et al., 2011; Ye & Doermann, 2012) with 768 values (bins) to represent the degraded image. Then, we compute the pairwise similarities of all degraded images. Next, we implement spectral clustering based on the computed similarity matrix to generate 100 cluster centers. The degradation parameters of the cluster centers are then utilized to generate the distributional test set $\mathcal{D}^{\text{test}}$. We take Set14 (Zeyde et al., 2010) and DIV2K_val (Lim et al., 2017) to construct the test sets for systematic evaluation, denoted as Set14-SE and DIV2K_val-SE, respectively.

**Establishing the acceptance and excellence lines.** We use the 100 representative degradation parameters to synthesize 100 training datasets based on DIV2K. In the case of MSE-based real-SR methods, we utilize a variant of FSRCNN (Dong et al., 2016), referred to as FSRCNN-mz, to train a collection of 100 non-blind SR models, which serve as acceptance line. Concurrently, we employ SRResNet (Ledig et al., 2017), following an identical procedure, as the excellence line [1]. The models within the model zoo are initially pre-trained under the real-SR setting. Subsequently, they undergo a fine-tuning process consisting of a total of $2 \times 10^5$ iterations. The Adam (Kingma & Ba, 2014) optimizer with $\beta_1 = 0.9$ and $\beta_2 = 0.99$ is used for training. The initial learning rate is $2 \times 10^{-4}$. We adopt L1 loss to optimize the networks. Regarding GAN-based SR methods, we adopt the widely recognized RealESRGAN (Wang et al., 2021b) as our acceptance line. Concurrently, we consider the state-of-the-art RealHATGAN (Chen et al., 2023b;a) as our excellence line. We utilize the officially released models for our experiments.

---

[1]Both the two lines and the distributed test set will be released.

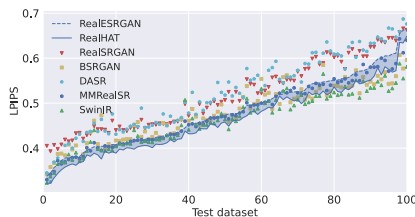

Figure 6: **Visualization of distributed performance** in LPIPS for GAN-based real-SR methods on Set14-SE.

Table 2: **Benchmark results and ranking of GAN-based real-SR methods** in LPIPS by our proposed SEAL. The subscript denotes the rank order. $\times$ means the model fails in a majority of degradation cases.

| Models | LPIPS $\downarrow$ | $AR\uparrow$ | $RPR_I\downarrow$ | $RPR_A\uparrow$ | $RPR_U\uparrow$ | Rank |
|---|---|---|---|---|---|---|
| ESRGAN | $0.6224_{(6)}$ | $0.00_{(\times)}$ | 0.01 | 0.00 | 0.03 | $\times$ |
| RealSRGAN | $0.5172_{(5)}$ | $0.01_{(\times)}$ | 0.10 | 0.53 | 0.14 | $\times$ |
| DASR | $0.5230_{(4)}$ | $0.02_{(\times)}$ | 0.13 | 0.61 | 0.12 | $\times$ |
| BSRGAN | $0.4810_{(3)}$ | $0.44_{(3)}$ | $0.40_{(3)}$ | $0.72_{(1)}$ | $0.28_{(3)}$ | 3 |
| MMRealSR | $0.4770_{(2)}$ | $0.80_{(2)}$ | $0.08_{(1)}$ | $0.57_{(3)}$ | $0.41_{(1)}$ | 1 |
| SwinIR | $0.4656_{(1)}$ | $0.81_{(1)}$ | $0.24_{(2)}$ | $0.71_{(2)}$ | $0.31_{(2)}$ | 2 |

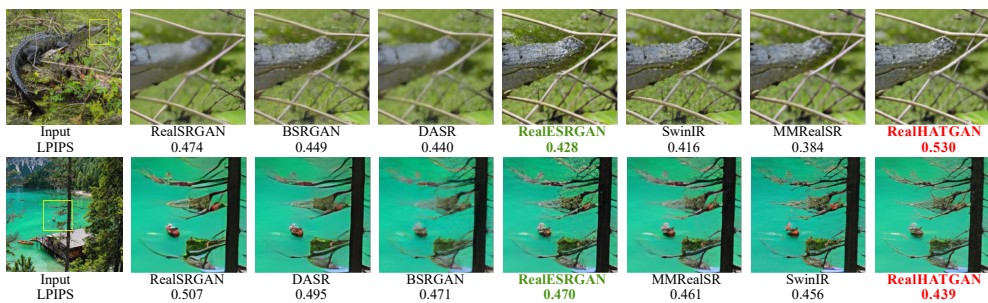

Figure 7: **Visual results of GAN-based real-SR methods** with the acceptance line RealESRGAN and excellence line RealHATGAN. It is best viewed in color.

## 5.2 BENCHMARKING EXISTING MSE-BASED AND GAN-BASED REAL-SR METHODS

We utilize the proposed SEAL to evaluate the performance of existing MSE-based real-SR methods, including DASR (Wang et al., 2021a), BSRNet (Zhang et al., 2021), SwinIR (Liang et al., 2021a), RealESRNet (Wang et al., 2021b), RDSR (Kong et al., 2022), and RealESRNet-GD (Zhang et al., 2022). Furthermore, we benchmark GAN-based real-SR methods such as ESRGAN (Wang et al., 2018), DASR (Liang et al., 2022), BSRGAN (Zhang et al., 2021), MMRealSR (Mou et al., 2022), SwinIR (Liang et al., 2021a). We also modify SRGAN (Ledig et al., 2017) to achieve RealSRGAN under the RealESRGAN training setting.

**The visualization of the distributed performance offers a comprehensive insight into real-SR performance.** Fig. 4 illustrates the distribution performance for MSE-based real-SR methods, using PSNR as the IQA metric. On the other hand, Fig. 6 depicts the distribution performance for GAN-based real-SR methods, with LPIPS as the metric. Both visualizations are generated using our proposed SE test set. The SE test sets are arranged in ascending order based on the PSNR values of the acceptance line output. Test sets with lower numbers represent more challenging cases. It's noticeable that there are a few degradation cases that fall significantly below the acceptance line in Fig. 4. Interestingly, real-SR methods seem to perform better on more challenging degradation cases. This is evident in test datasets 0-20 in Fig. 4 and 80-100 in Fig. 6.

**The coarse-to-fine evaluation protocol offers a systematic ranking.** In Tab. 1 and Tab. 2, the real-SR models with $AR$ below 0.25 are excluded from the ranking due to their low acceptance rates. For the real-SR models with $AR > 0.25$, a step-by-step ranking is performed based on $\{AR, RRP_I, RPR_A, RPR_U\}$, with thresholds $\{0.02, 0.02, 0.05, 0.05\}$ respectively. If the difference in the current metric exceeds the threshold, the metric is used to represent the overall ranking. Otherwise, the next metric is considered. From our proposed SEAL evaluation, we can make several observations: (1) **Some existing methods fail on the majority of degradation cases.** The $AR$ values of some existing methods are below 0.5, as shown in Tab. 1 and Tab. 2. For instance, most MSE-based real-SR models can not even outperform the small network (i.e., FSRCNN-mz) in most degradation cases. (2) **Our SEAL is capable of ranking existing methods across various dimensions, such as robustness, denoted by $RPR_I$ and performance bound indicated by $RPR_A$.** In Tab. 2, the metric learning based MMRealSR achieves significant robustness ($RPR_I$ 0.08) compared with

the transformer-based SwinIR ($RPR_I$ 0.24). Therefore, under our current coarse-to-fine evaluation protocol, MMRealSR is ranked in the first place. Interestingly, we observed that SwinIR achieves a higher $RPR_A$ at the same $AR$ level. If the user prioritizes the performance of acceptance cases, SwinIR would be a better choice. Consequently, we can also flexibly set $RPR_A$ as the first finer metric. In this way, SwinIR would take the first place. (3) **The acceptance line serves as a useful reference line for visual comparison**. Visual results are presented in Fig. 5 and Fig. 7. It's evident that the visual results of the acceptance line can serve as a basic need for image quality, while the visual results of the excellence line represent the upper bound of image quality under the current evaluation protocol. The visuals below the acceptance line clearly exhibit unacceptable visual effects, including blurring (as seen in the *crocodile* results of RealSRGAN and DASR in Fig. 7), over-sharpening (as seen in the *text* results of RealESRNet in Fig. 5), and other artifacts. Notably, our SEAL can flexibly use new reference lines for future needs.

## 5.3 COMPARISON WITH THE CONVENTIONAL EVALUATION

Here, we compare our SEAL with the conventional strategy (Zhang et al., 2021; Liang et al., 2022) used for evaluating real-SR models.

**Randomly generated multiple synthetic test sets fail to establish a clear ranking with distinct differences.** We randomly sample 100 degradation cases and add them to Set14 to obtain 100 test sets (Set14-Random). Tab. 3 shows that 1) the mean and standard deviations (std) of PSNR obtained on the two Set14-Random100 datasets show significant inconsistency, demonstrating the presence of high randomness and variability in the sampled degradation cases. 2) On our Set14-SE (formed with the 100 representative cases), the means and stds of the compared methods are very close, making it hard to establish a clear ranking with distinct differences among the methods. In contrast, our SEAL offers a definitive ranking of these methods based on their $AR$ scores, offering a new systematic evaluation view.

Table 3: **Comparison with multiple synthetic test sets** on mean and standard deviations.

| PSNR↑ | Set14-Random100 (#1) mean↑ | std↓ | Set14-Random100 (#2) mean↑ | std↓ | Set14-SE mean | std | $AR$↑ | $RPR_I$↓ | $RPR_A$↑ | $RPR_U$↑ | rank |
|---|---|---|---|---|---|---|---|---|---|---|---|
| BSRNet | 23.39(3) | 1.56(2) | 22.98(1) | 1.64(1) | 22.77(2) | 1.65(1) | 0.59(1) | 0.42(4) | 0.72(2) | 0.27(4) | 1 |
| RealESRNet-GD | 23.72(1) | 1.64(4) | 22.98(1) | 1.95(4) | 22.82(1) | 1.83(4) | 0.43(2) | 0.37(3) | 0.74(1) | 0.33(1) | 2 |
| SwinIR | 23.25(4) | 1.62(3) | 22.79(4) | 1.69(2) | 22.61(4) | 1.69(2) | 0.41(3) | 0.24(1) | 0.58(4) | 0.29(2) | 3 |
| RealESRNet | 23.54(2) | 1.55(1) | 22.80(3) | 1.83(3) | 22.67(3) | 1.73(3) | 0.27(4) | 0.28(2) | 0.63(3) | 0.28(3) | 4 |

**The utilization of a randomly generated synthetic test set may lead to misleading outcomes.** Following Zhang et al. (2021); Liang et al. (2022), we randomly add degradations to images in the DIV2K (Agustsson & Timofte, 2017) validation set to construct a single real-DIV2K_val set. Tab. 4 shows that RealESRNet achieves a higher average PSNR than BSRNet (24.93dB vs. 24.77dB) on real-DIV2K_val, leading to the misleading impression that RealESRNet is superior than BSRNet. However, our SEAL framework leads to a contrary conclusion. BSRNet obtains much higher PSNR (24.74dB vs. 24.43dB) and $AR$ value (0.55 vs. 0.15) than RealESRNet on our DIV2K_val-SE, illustrating that the former outperforms the latter in most representative degradation cases.

Table 4: **Comparison with the single synthetic test set.** Under our SEAL framework, BSRNet is ranked first, contrary to the results obtained by the conventional method. PSNR-SE (Eq. 4) denotes the average PSNR on our DIV2K_val-SE.

| | PSNR | Rank | PSNR-SE | $AR$↑ | $RPR_I$↓ | $RPR_A$↑ | $RPR_U$↑ | Rank (ours) |
|---|---|---|---|---|---|---|---|---|
| BSRNet | 24.77 | 2 | 24.74 | 0.55(1) | 0.36 | 0.65 | 0.26 | 1 |
| RealESRNet | 24.93 | 1 | 24.43 | 0.15(2) | 0.33 | 0.59 | 0.18 | 2 |

## 5.4 ABLATION STUDIES AND ANALYSIS

In this section, we first conduct ablation studies on several factors that affect spectral clustering, including the number of sampled degradations, similarity metrics, and the number of clusters. Then, we study the stability of degradation clustering for real-SR evaluation.

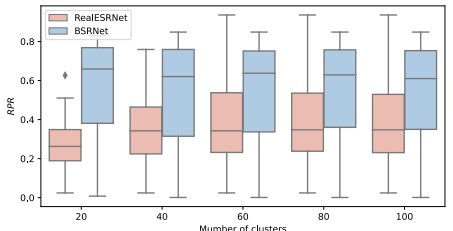 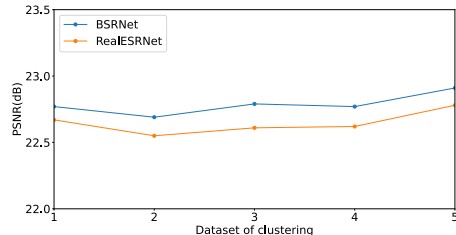

Figure 8: **Effect of the number of clusters** on $RPR$ value of Set14-SE.

Figure 9: **Effect of the dataset used for clustering** on average PSNR of Set14-SE.

**Number of sampled degradations.** To assess the effect of the number of sampled degradations, we randomly generate four datasets, each containing 500, 1000, 5000, and 10000 degradation samples, respectively. We compute the variance of the similarity matrices for each of these datasets, which are 8.32, 8.45, 8.68, and 8.71, respectively. The observation indicates that the change in variance is not significant when the number of samples increases from 5000 to 10000. This observation suggests that a sample size of 5000 random degradations sufficiently represents the degradation space. However, to ensure the highest possible accuracy in our results, we opted to use 10000 degradation samples for the clustering process.

**Choice of similarity metric.** We compare different metrics that are used to compute the similarity matrix for degradation clustering, including MSE, SSIM, and histogram similarity. In Tab. 5, the purity accuracy of the clustering results using MSE or SSIM is significantly lower than that with histogram similarity, especially when noise is considered. Thus, we adopt histogram similarity as the similarity metric.

Table 5: **The purity accuracy** of the clustering results with different similarity metrics on Blur100, Noise100, and BN100 datasets.

|  | Range | $K$ | MSE | SSIM | Histogram |
|---|---|---|---|---|---|
| Blur100 | 0.1 - 4 | 4 | 78.2% | 78.2% | **80.2%** |
| Noise100 | 1 - 40 | 4 | 39.6% | 34.6% | **80.2%** |
| Blur100 + Noise100 | - | 8 | 51.7% | 58.2% | **80.5%** |

**Choice of the number of clusters.** Our goal is to generate as many representative classes as possible while maintaining the clustering quality so that the class centers can serve as representative cases. The results in Fig. 8 show the performance of RealESRNet and BSRNet becomes stable as $k$ approaches 100, with minimal variations observed for $k = 60, 80, 100$. Therefore, to achieve a more comprehensive assessment and strike a balance between clustering quality and time cost, we set $k = 100$ without further increasing its value. In the appendix, we have included the quantitative results of the silhouette score (Rousseeuw, 1987), a metric commonly employed to evaluate the quality of clusters.

**Stability of degradation clustering for real-SR evaluation.** In Fig. 9, we study the stability of degradation clustering by using different images as a reference for evaluation. Beyond the *Lenna* image, our study incorporated four additional images—specifically, *Baboon*, *Barbara*, *Flowers*, and *Zebra*—from the Set14 dataset. These images were employed as Ground Truth images in the construction of the clustering dataset, adhering to the same degradation clustering process. Despite using different reference images, our results show that the average PSNR of BSRNet is consistently higher than that of RealESRNet by more than 0.1dB, indicating that our degradation clustering method exhibits excellent stability for real-SR evaluation.

## 6 CONCLUSIONS

In this work, we have developed a new evaluation framework for a fair and comprehensive evaluation of real-SR models. We first use a clustering-based approach to model a large degradation space and design two new evaluation metrics, $AR$ and $RPR$, to comparatively assess real-SR models on representative degradation cases. Then, we benchmark existing real-SR methods with the proposed evaluation protocol and present new observations and insights. Finally, extensive ablation studies are conducted on the degradation clustering. We have demonstrated the effectiveness and generality of SEAL via extensive experiments and analysis.

# 7 ACKNOWLEDGMENTS

This work was supported in part by the National Natural Science Foundation of China under Grant (62276251,62272450).

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

# A    ADAPTABILITY OF OUR SEAL FRAMEWORK

We contend that the components of our SEAL framework are highly adaptable to user preferences. For instance, users have the option to choose a different reference line to visualize distributed performance, reorganize the SE test sets into new groups, and utilize any IQA metrics for evaluation.

## A.1    INCORPORATING NEW IQA METRICS

To illustrate the adaptability of our SEAL framework, we have opted for SSIM as the IQA metric to perform a comprehensive evaluation of real-SR methods. As depicted in Tab. 6, RealESRNet surpasses other methods in terms of $AR$, an outcome that can be credited to the use of sharpened Ground-Truth images. It is significant that RealESRNet and SwinIR exhibit remarkable stability, as evidenced by their $RPR_I$ values. Furthermore, our findings indicate that SwinIR attains the highest $RPR_A$ value, implying that transformer-based networks favor acceptance degradation cases. As evidenced by these observations, our proposed evaluation framework displays considerable adaptability. It accommodates various IQA metrics to systematically evaluate real-SR methods from diverse angles, such as reconstruction capability (PSNR) and structural similarity (SSIM).

Table 6: Results and ranking of different methods in SSIM by our SEAL framework. The subscript denotes the rank order. $_\times$ represents a failed SR model in a large degradation space.

| Set14-SE | $AR\uparrow$ | $RPR_I\downarrow$ | $RPR_A\uparrow$ | $RPR_U\uparrow$ | Rank |
|---|---|---|---|---|---|
| SRResNet | $0.00_{(\times)}$ | $0.04$ | $0.00$ | $0.04$ | $\times$ |
| DASR | $0.00_{(\times)}$ | $0.03$ | $0.00$ | $0.04$ | $\times$ |
| BSRNet | $0.76_{(3)}$ | $0.27_{(5)}$ | $0.70_{(2)}$ | $0.36_{(4)}$ | 3 |
| RealESRNet | $0.91_{(1)}$ | $0.16_{(1)}$ | $0.67_{(3)}$ | $0.43_{(1)}$ | 1 |
| RDSR | $0.32_{(5)}$ | $0.22_{(3)}$ | $0.59_{(5)}$ | $0.33_{(5)}$ | 5 |
| RealESRNet-GD | $0.69_{(4)}$ | $0.26_{(4)}$ | $0.67_{(3)}$ | $0.39_{(2)}$ | 4 |
| SwinIR | $0.84_{(2)}$ | $0.17_{(2)}$ | $0.72_{(1)}$ | $0.38_{(3)}$ | 2 |

## A.2    USING MULTIPLE REFERENCE IMAGES FOR DEGRADATION CLUSTERING

To illustrate the adaptability of the component in our SEAL framework, we have incorporated a new experiment by utilizing the 5 reference images in Figure 9 for degradation clustering. To accomplish this, we computed the average of the similarity matrices induced by these images. We used the newly identified representative degradation cases to rank GAN-based methods and provided the results in Tab. 7. Notably, we have observed that this adjustment has led to a more conclusive ranking compared to the results obtained when using a single image (Tab. 2). In Tab. 2, we observed that SwinIR and MMRealSR had a very close difference of only 0.01 in the $AR$ metric. Consequently, their ranks needed to be determined using finer metrics such as $RPR_I$. However, in Table 7, we found that SwinIR ($AR$: 0.86) and MMRealSR ($AR$: 0.75) could be easily ranked based on the $AR$ metric alone. This suggests that the degradation cases identified by combining the 5 images may indeed be more representative. It also emphasizes the potential benefits of applying enhanced clustering algorithms, which can further enhance the stability and representativeness of our SEAL framework.

Table 7: Benchmark results and ranking of GAN-based real-SR methods in LPIPS by our proposed SEAL with five reference images. The subscript denotes the rank order. × means the model fails in a majority of degradation cases.

| Models | LPIPS-SE$\downarrow$ | $AR\uparrow$ | $RPR_I\downarrow$ | $RPR_A\uparrow$ | $RPR_U\uparrow$ | Rank |
|---|---|---|---|---|---|---|
| ESRGAN | $0.6152_{(6)}$ | $0.01_{(\times)}$ | $0.01$ | $0.73$ | $0.03$ | $\times$ |
| RealSRGAN | $0.5180_{(5)}$ | $0.02_{(\times)}$ | $0.16$ | $0.55$ | $0.15$ | $\times$ |
| DASR | $0.5228_{(4)}$ | $0.04_{(\times)}$ | $0.13$ | $0.59$ | $0.13$ | $\times$ |
| BSRGAN | $0.4809_{(3)}$ | $0.52_{(3)}$ | $0.33_{(3)}$ | $0.69_{(2)}$ | $0.29_{(2)}$ | 3 |
| MMRealSR | $0.4777_{(2)}$ | $0.75_{(2)}$ | $0.10_{(1)}$ | $0.59_{(3)}$ | $0.41_{(1)}$ | 2 |
| SwinIR | $0.4672_{(1)}$ | $0.86_{(1)}$ | $0.19_{(2)}$ | $0.71_{(1)}$ | $0.28_{(3)}$ | 1 |

## A.3 USER-CUSTOMIZED SE TEST SETS

In order to accommodate varying user preferences, such as the analysis of the quantitative performance of IQA metrics, the SE test sets are organized in ascending order based on the PSNR values of the FSRCNN-mz output. These sets are then partitioned into five groups of equal size. Group 1 encompasses the most challenging cases, while Group 5 includes the least challenging ones. As shown in Table 8, the average RPR value of BSRNet closely matches that of RealESRNet-GD. However, there is a variation in their performance across different groups. RealESRNet-GD outperforms in groups {3, 4, 5}, whereas BSRNet takes the lead in groups {1, 2}.

Table 8: $RPR$ value of different methods on Set14-SE in PSNR. Blue: better than FSRCNN-mz.

| Model | SRResNet | DASR | BSRNet | RealESRNet | RDSR | RealESRNet-GD | SwinIR |
|---|---|---|---|---|---|---|---|
| Group 1 | 0.03 | 0.03 | **0.64** | 0.37 | 0.26 | 0.34 | 0.48 |
| Group 2 | 0.02 | 0.02 | **0.60** | 0.37 | 0.21 | 0.45 | 0.43 |
| Group 3 | 0.07 | 0.07 | 0.51 | 0.42 | 0.31 | **0.57** | 0.40 |
| Group 4 | 0.02 | 0.01 | 0.37 | 0.40 | 0.29 | **0.68** | 0.27 |
| Group 5 | 0.03 | 0.03 | 0.44 | 0.36 | 0.17 | **0.55** | 0.36 |
| Average | 0.03 | 0.03 | 0.51 | 0.38 | 0.25 | **0.52** | 0.39 |

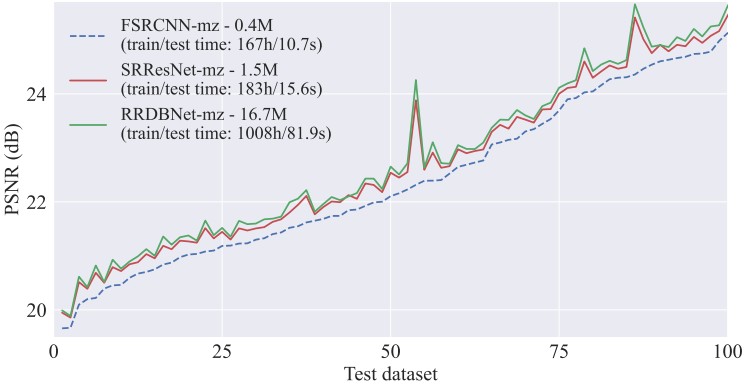

Figure 10: **Comparison of network structures** for the acceptance and excellence lines.

## A.4 EXTENSION ON ACCEPTANCE LINE AND EXCELLENCE LINE.

For the acceptance line, we hope it can represent an acceptable lower bound of performance with good discrimination for different models. Concretely, the acceptance line cannot be so high that $AR$ of most methods cannot exceed 0, nor can it be so low that $AR$ can easily reach 1.0. FSRCNN (Dong et al., 2016) is a small network (0.4M Params.) while it can distinguish the performance difference well, as shown in Tab. 1. Therefore, we choose FSRCNN-mz as the acceptance line.

For the excellence line, we compare the networks of SRResNet (Ledig et al., 2017) (1.5M Params.) and RRDBNet (Wang et al., 2018) (16.7M Params.). In Fig. 10, we observe that SRResNet-mz and FSRCNN-mz can already distinguish the performance difference. Although RRDBNet-mz exhibits a slight performance improvement, it comes at the expense of increased training and testing time, far surpassing those of other models. Considering the trade-off between performance and time costs, we choose SRResNet-mz as the excellence line. Nonetheless, we emphasize that our rationale for choosing these two lines is that they can well differentiate the methods for comparison. Note that the two lines can be changed flexibly to meet specific requirements of other scenarios.

## A.5 TIME COST OF OUR EVALUATION FRAMEWORK

We provide information on time cost in Tab. 9. As expected, our approach does result in an increase in inference time, which scales linearly with the number of identified representative degradation cases (e.g., 100 in our experiments). However, since the inference time remains within acceptable limits, we believe this is a worthwhile tradeoff between evaluation efficiency and quality.

Table 9: Comparison of the time cost between the conventional evaluation method and our approach using RRDBNet (Zhang et al., 2021; Wang et al., 2021b).

|  | inference time [s] | PSNR run-time [s] | $AR, RPR$ run-time [s] |
|---|---|---|---|
| Set14 | 4.22 | 0.52 | 0.013 |
| Set14-SE (ours) | 382.74 | 49.47 | 0.013 |

Table 10: **Basic strategies for model comparison.** We use three basic strategies to compare the overall performance of real-SR models. The real-SR models are trained on: (1) different network structures, (2) different training datasets, and (3) the RealESRGAN degradation model with different gate probability as proposed in (Zhang et al., 2022).

|  |  | $AR\uparrow$ | $RPR_I\downarrow$ | $RPR_A\uparrow$ | $RPR_U\uparrow$ | Rank |
|---|---|---|---|---|---|---|
| Network (Parameter [M]) | SRResNet (1.5) | $0.12_{(\times)}$ | 0.20 | 0.63 | 0.26 | $\times$ |
|  | RCAN (15.6) | $0.37_{(2)}$ | $\mathbf{0.15}_{(1)}$ | $0.62_{(2)}$ | $0.39_{(2)}$ | 2 |
|  | RRDBNet (16.7) | $0.37_{(2)}$ | $0.33_{(3)}$ | $0.68_{(1)}$ | $0.32_{(3)}$ | 3 |
|  | SwinIR (11.9) | $\mathbf{0.67}_{(1)}$ | $\mathbf{0.15}_{(1)}$ | $0.62_{(2)}$ | $0.41_{(1)}$ | 1 |
| Training dataset | DIV2K | $0.32_{(3)}$ | $0.25_{(3)}$ | $0.64_{(2)}$ | $0.33_{(3)}$ | 3 |
|  | DF2K | $0.43_{(2)}$ | $0.24_{(2)}$ | $0.67_{(1)}$ | $0.39_{(2)}$ | 2 |
|  | ImageNet | $\mathbf{0.63}_{(1)}$ | $0.22_{(1)}$ | $0.67_{(1)}$ | $0.41_{(1)}$ | 1 |
| Gate probability | 1.00 | $0.37_{(4)}$ | $0.33_{(1)}$ | $0.68_{(3)}$ | $0.32_{(2)}$ | 3 |
|  | 0.75 | $\mathbf{0.44}_{(1)}$ | $0.35_{(2)}$ | $0.69_{(2)}$ | $0.34_{(1)}$ | 1 |
|  | 0.50 | $0.43_{(2)}$ | $0.35_{(2)}$ | $0.70_{(1)}$ | $0.31_{(3)}$ | 1 |
|  | 0.25 | $0.40_{(3)}$ | $0.43_{(4)}$ | $0.66_{(4)}$ | $0.21_{(4)}$ | 4 |
| BSRNet (SOTA) |  | $0.59_{(2)}$ | $0.42_{(2)}$ | $0.72_{(1)}$ | $0.27_{(2)}$ | 2 |
| SwinIR-GD-I (Ours) |  | $\mathbf{0.85}_{(1)}$ | $\mathbf{0.25}_{(1)}$ | $\mathbf{0.72}_{(1)}$ | $\mathbf{0.40}_{(1)}$ | 1 |

# B DEVELOPING NEW STRONG REAL-SR MODELS

According to the evaluation results by our framework, as shown in Tab. 10, we can improve the real-SR performance in **three** aspects to develop a stronger real-SR model: **1)** A powerful backbone is vital for overall performance. We can observe that SwinIR obtains the highest $AR$ and the lowest $RPR_I$. **2)** Using a large-scale dataset (i.e., ImageNet (Deng et al., 2009)) can also greatly improve the real-SR performance. **3)** A degradation model with the appropriate distribution (i.e., gate probability: 0.75 (Zhang et al., 2022)) also has a non-negligible impact on the real-SR performance. Based on these observations, we use SwinIR as the backbone to train a new strong real-SR model on ImageNet with a high-order gate degradation (GD) model (gate probability: 0.75), denoted as SwinIR-GD-I. The evaluation results in Tab. 10 show that SwinIR-GD-I obtain a significant improvement over the SOTA performance of BSRNet. Fig. 11 shows that the visual results of SwinIR-GD-I are obviously better than BSRNet and SwinIR. We believe our framework would inspire more powerful real-SR methods in the future.

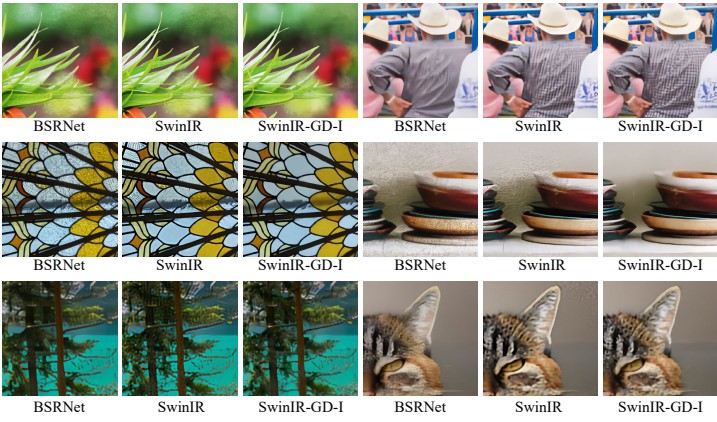

Figure 11: Visual results of the proposed baseline SwinIR-GD-I and other real-SR methods.

## C  LIMITATIONS OF THE CONVENTIONAL EVALUATION METHOD

To further demonstrate the limitations of the traditional evaluation method, we generated 20 random test sets for comprehensive analysis. Using a large degradation model, we randomly apply degradations to the images from the DIV2K_val dataset. As shown in Fig. 12, each bar represents the PSNR difference between BSRNet and RealESRNet on a single test set. Across these 20 test sets, BSRNet and RealESRNet each outperform the other in approximately half of the cases, with their performance gap ranging from -0.22 to 0.18. It is difficult to definitively determine which method is superior. This observation further highlights the inadequacy of using randomized test sets for evaluation.

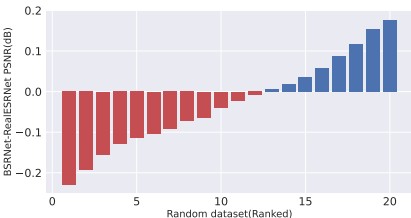

Figure 12: Results of PSNR distance between BSR-Net and RealESRNet on 20 randomly generated test sets, sorted in ascending order.

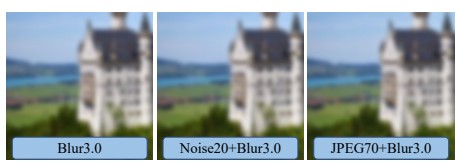

Figure 13: Similar visual effects of different degradation combinations.

## D  DETAILS OF DEGRADATION CLUSTERING

### D.1  SPECTRAL CLUSTERING

We use the shuffled degradation model of BSRGAN (Zhang et al., 2021) and the high-order degradation model of RealESRGAN (Wang et al., 2021b) to construct a large degradation space. The degraded images are generated by the shuffled (Zhang et al., 2021) and high-order (Wang et al., 2021b) degradation models with probabilities of $\{0.5, 0.5\}$. The degradation types mainly consist of 1) various types of Gaussian blur; 2) commonly-used noise: Gaussian, Poisson, and Speckle noise with gray and color scale; 3) multiple resize strategy: area, bilinear and bicubic; 4) JPEG noise.

---

**Algorithm 1** Image degradation clustering

---

**Input:** Similarity matrix $S \in \mathbb{R}^{n \times n}$, number $k$ of clusters to construct.
1: Compute Adjacency Matrix $W$ and Degree Matrix $D$.
2: Compute Laplacian Matrix $L = D - W$.
3: Compute the first $K$ eigenvectors $u_1, ..., u_k$ of $L$.
4: Let $U \in \mathbb{R}^{n \times k}$ be the matrix containing the vectors $u_1, ..., u_k$ as columns.
5: For $i = 1, ..., n$, let $y_i \in \mathbb{R}^k$ be the vector corresponding to the $i$-th row of $U$.
6: Cluster the points $(y_i)_{i=1,...,n}$ in $\mathbb{R}^k$ with the k-means algorithm into clusters $C_1, ..., C_k$
**Output:** Cluster centers $c_1, ..., c_K$ with $c_i \in C_i$.

---

We use spectral clustering to group the degraded images $(x)$ due to its effectiveness and flexibility in finding arbitrarily shaped clusters. First, we use the RGB histogram $(h)$ (Tang et al., 2011; Ye & Doermann, 2012) with 768 values (bins) as the image feature to calculate the similarity $s_{ij} = L_1(h(x_i), h(x_j))$. The histograms of R, G, and B are generated separately and then concatenated into a single vector to represent the degraded image. The similarity matrix is defined as a symmetric matrix $S$, where $s_{ij}$ represents a measure of the similarity between data points $x$ with indices $i$ and $j$ for $n$ data points. We execute Algo. 1 step by step to obtain the degradation parameter of cluster centers $\mathcal{D} = \{c_1, c_2, ..., c_K\}$. Then, we use the degradation parameter of cluster centers as the representative degradations to construct the systematic set.

### D.2    SIMILARITY METRICS

In this section, we provide more experimental details for the *Sec. similarity metrics* in the main paper. To select an appropriate similarity metric, we create two datasets with simple degradation types – Gaussian blur with a range of [0.1, 4.0] and Gaussian noise [1, 40]. We use the image *lenna* in Set14 (Zeyde et al., 2010) as our Ground-Truth image. Firstly, we generate 100 low-quality images named Blur100 by applying Gaussian blur within a range of [0.1, 4.0]. Each cluster is assigned a label based on the degradation intensity. We label the low-quality images with {[0.1, 1.0], [1.0, 2.0], [2.0, 3.0], [3.0, 4.0]} as {1, 2, 3, 4} respectively. Similarly, we generate 100 low-quality images named Noise100 by applying Gaussian noise within the range [1, 40], labeled as {1, 2, 3, 4} based on noise intensity.

To evaluate the effectiveness of the similarity metric, we combine Blur100 and Noise100 to produce BN100, which comprises 100 blurred images and 100 noised images. BN100 is labeled as {1, 2, 3, 4, 5, 6, 7, 8} using the same criteria as the previous datasets. Purity (Schütze et al., 2008) is an external criterion (similar to NMI, F measure (Schütze et al., 2008)) that assesses the alignment of the clustering outcome with the actual, known classes. It is defined as:

$$\text{Purity} = \frac{1}{N} \sum_{i=1}^{k} \max_{j} |c_i \cap t_j|, \tag{11}$$

where $N$ is the total number of data points, $\Omega = \{\omega_1, \omega_2, \ldots, \omega_K\}$ is the set of clusters and $\mathbb{C} = \{c_1, c_2, \ldots, c_J\}$ is the set of classes. To compute purity, each cluster is assigned to the class that appears most frequently within that cluster. The accuracy of this assignment is then evaluated by the number of correctly assigned points divided by the total number of points.

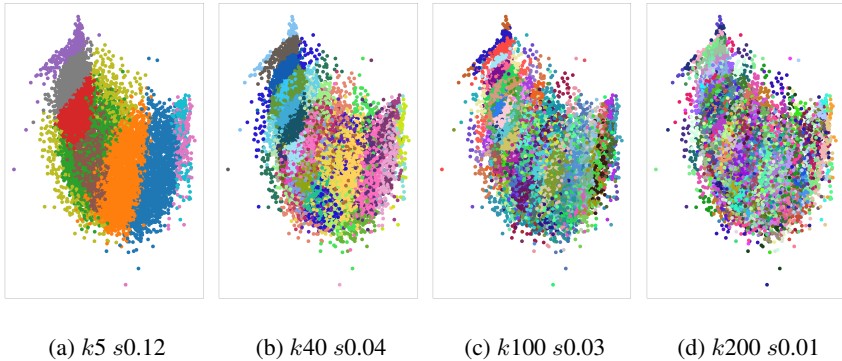

(a) $k5$ $s0.12$  (b) $k40$ $s0.04$  (c) $k100$ $s0.03$  (d) $k200$ $s0.01$

Figure 14: **Ablation of the number of clusters** $k$. $s$ denotes silhouette score.

### D.3    THE NUMBER OF CLUSTERS

To determine the number of clusters, we use silhouette scores (Rousseeuw, 1987) to measure the quality of the clusters. A higher silhouette score represents a better cluster, while the clustering result is acceptable if the silhouette score is greater than 0. As demonstrated in Fig. 14, the silhouette scores of k=40 and k=100 are very close, thus we utilize 100 clusters to find the representative cases.

## E    MORE EXPERIMENTAL RESULTS

### E.1    MORE VISUAL RESULTS ON REAL-SR METHODS

In this section, we further explore the effectiveness of our evaluation framework by providing additional qualitative results. We compare our proposed lines of acceptance and excellence with existing real-SR methods. The MSE-based methods that we consider include DASR (Wang et al., 2021a), BSRNet (Zhang et al., 2021), SwinIR (Liang et al., 2021a), RealESRNet (Wang et al., 2021b), RDSR (Kong et al., 2022), and RealESRNet-GD (Zhang et al., 2022). In Fig. 15, we use FSRCNN (green) to denote the acceptance line, and SRResNet (red) to represent the excellence line. Moving on to

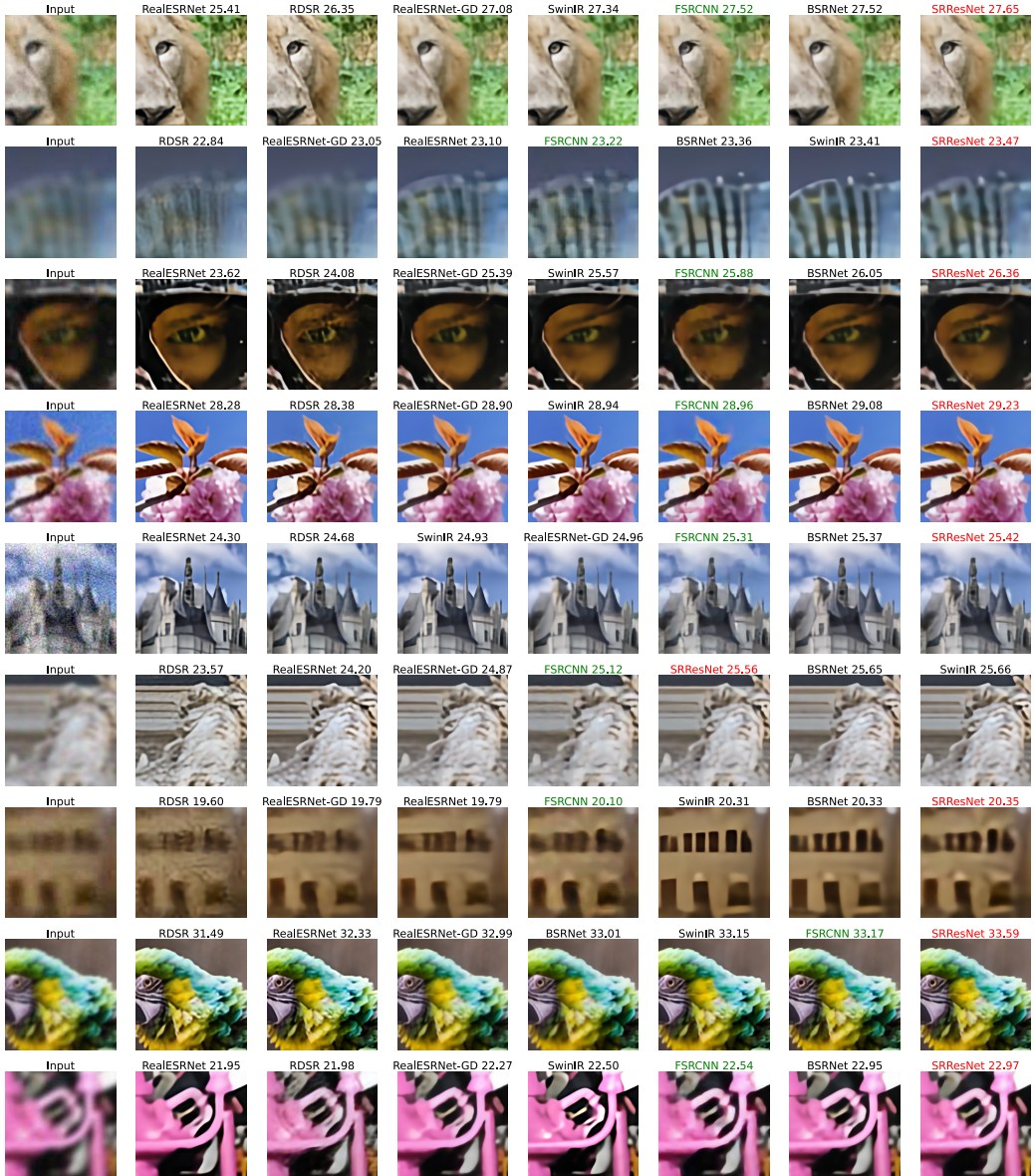

Figure 15: Visual results of MSE-based real-SR methods and the acceptance line FSRCNN and excellence line SRResNet with PSNR metric. It is best viewed in color.

the GAN-based methods, we include DASR (Liang et al., 2022), BSRGAN (Zhang et al., 2021), MMRealSR (Mou et al., 2022), SwinIR (Liang et al., 2021a) and RealSRGAN (Ledig et al., 2017). In Fig. 16, RealESRGAN (green) is used to denote the acceptance line, and RealHATGAN (red) is used to represent the excellence line. This comprehensive comparison provides a clear understanding of the performance of our proposed lines against the existing methods, thereby demonstrating the effectiveness of our evaluation framework.

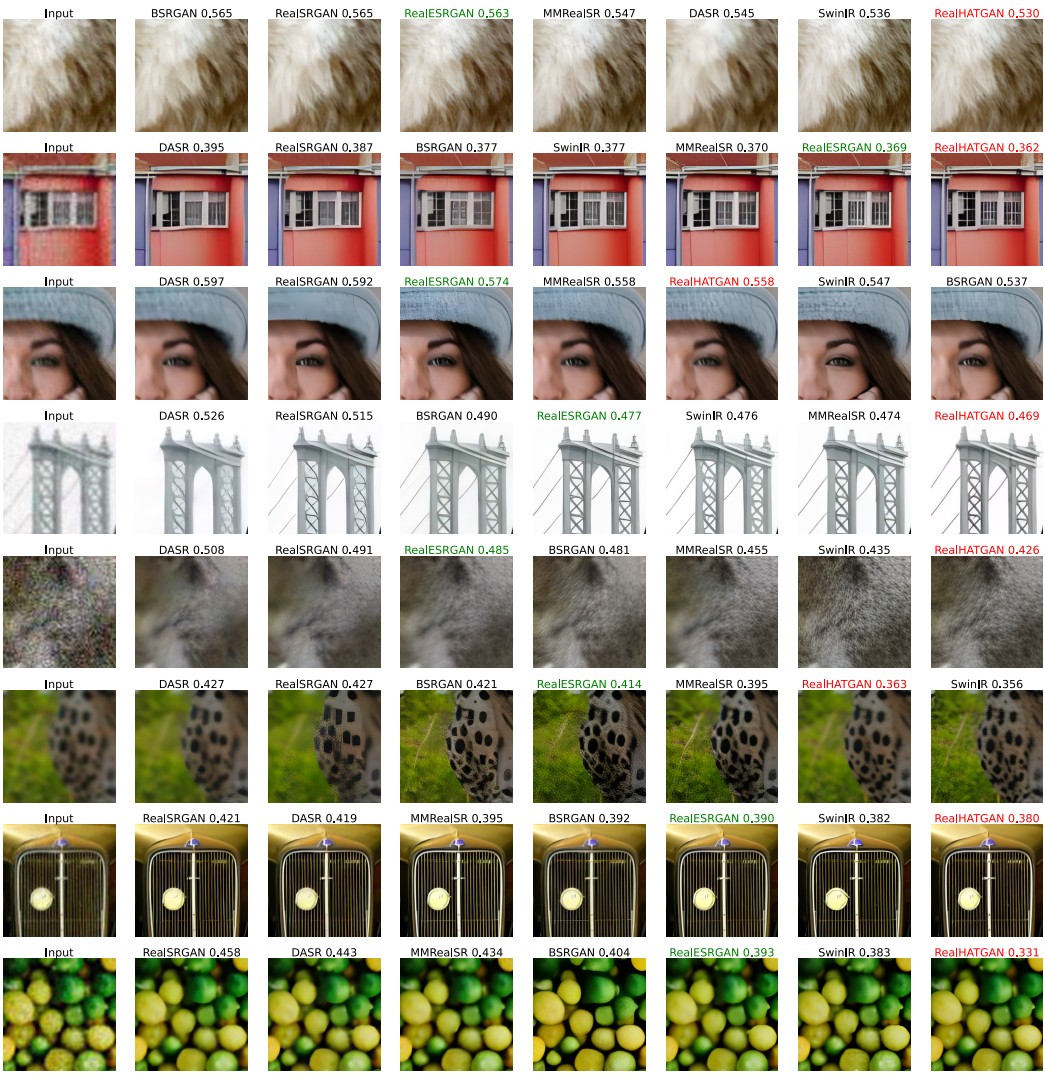

Figure 16: Visual results of GAN-based real-SR methods and the acceptance line RealESRGAN and excellence line RealHATGAN with LPIPS metric. It is best viewed in color.

## E.2 VISUAL RESULTS OF DEGRADATION CLUSTERING

We provide the visual comparison of samples both between and within clusters. In Fig. 17, we randomly selected five clusters out of a total of 100 clusters and chose six samples from each cluster. We observed that samples within each cluster exhibit similar degradation patterns, indicating they share a comparable level of restoration difficulty. In contrast, samples belonging to different clusters display remarkably distinct degradation patterns. In addition, we present the visualization of the degradation cluster centers in Fig. 18, Fig. 19, Fig. 20, Fig. 21 and Fig. 22. The cluster centers are sorted based on the PSNR value of the output of FSRCNN-mz. The results demonstrate that the degradation clustering algorithm performed as expected.

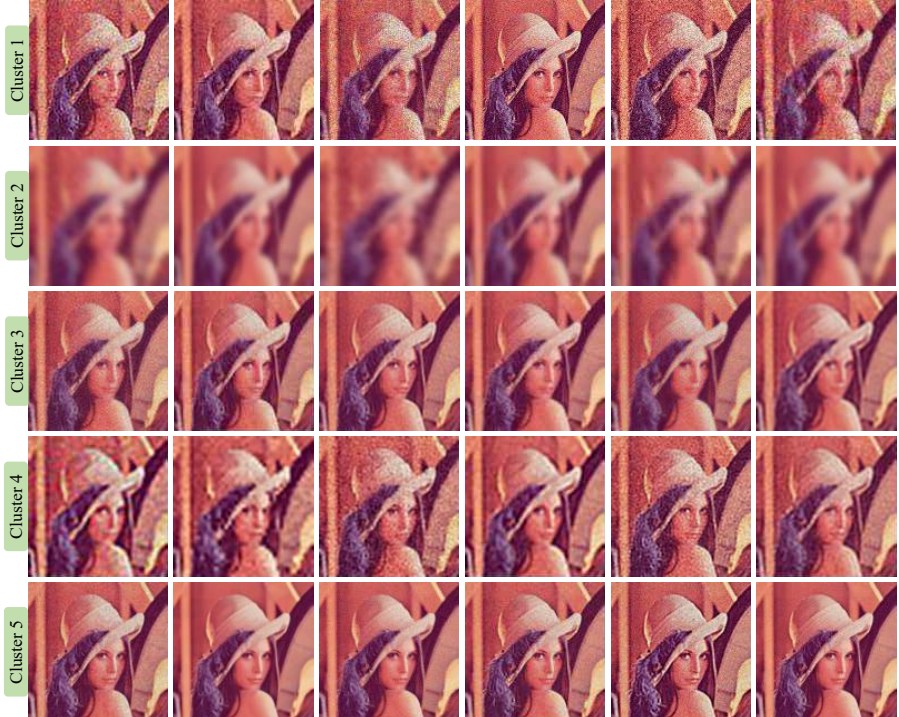

Figure 17: Visualization of samples from five randomly selected clusters. Best viewed in color.

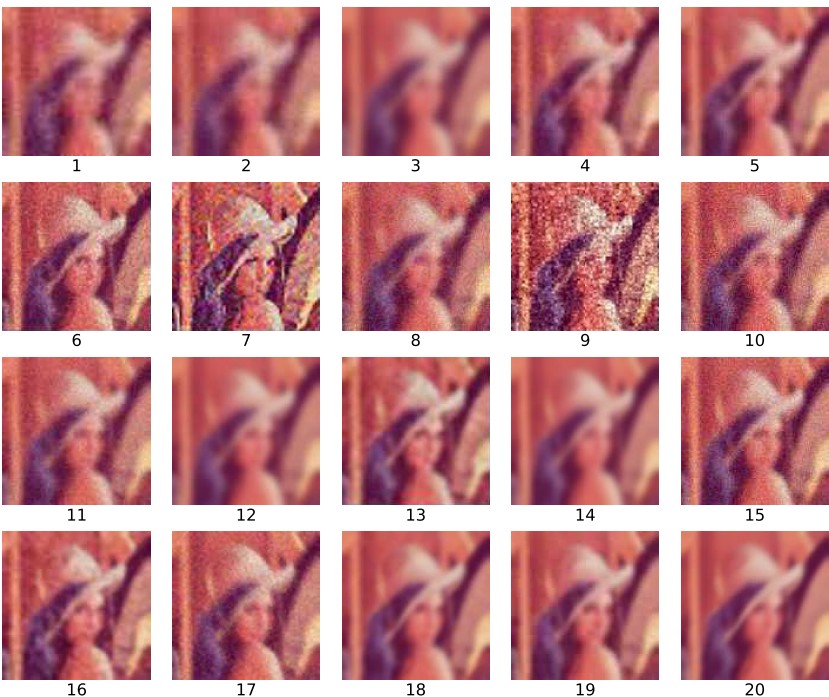

Figure 18: Visualization of the degradation cluster centers [1, 20], sorted by the PSNR of the output of FSRCNN-mz. Best viewed in color.

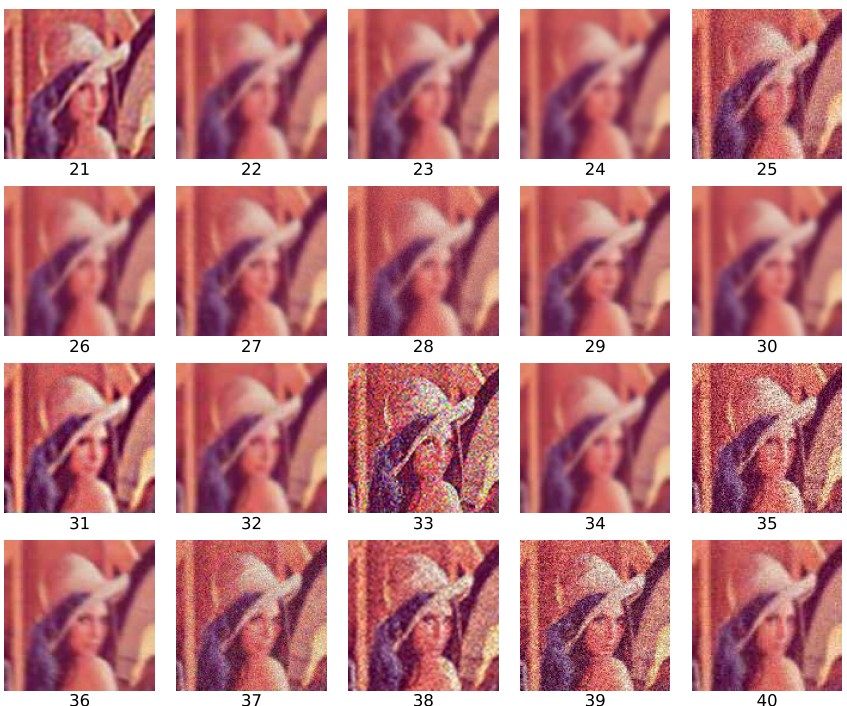

Figure 19: Visualization of the degradation cluster centers [21, 40], sorted by the PSNR of the output of FSRCNN-mz. Best viewed in color.

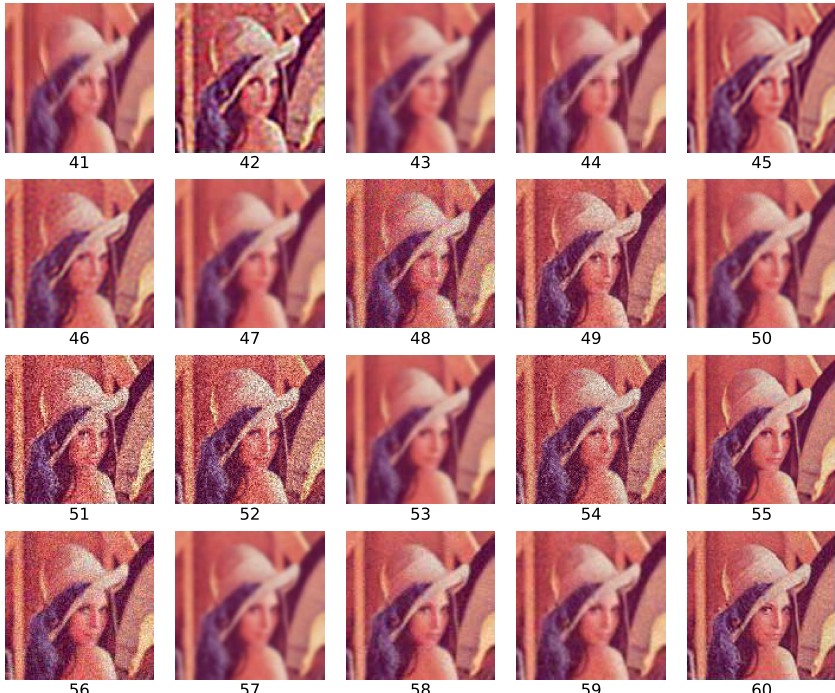

Figure 20: Visualization of the degradation cluster centers [41, 60], sorted by the PSNR of the output of FSRCNN-mz. Best viewed in color.

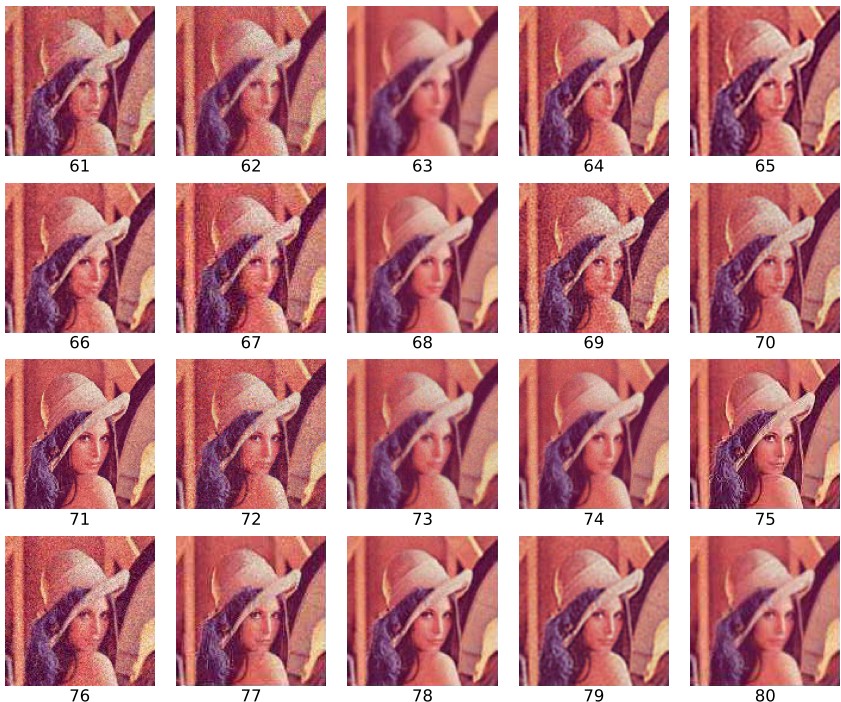

Figure 21: Visualization of the degradation cluster centers [61, 80], sorted by the PSNR of the output of FSRCNN-mz. Best viewed in color.

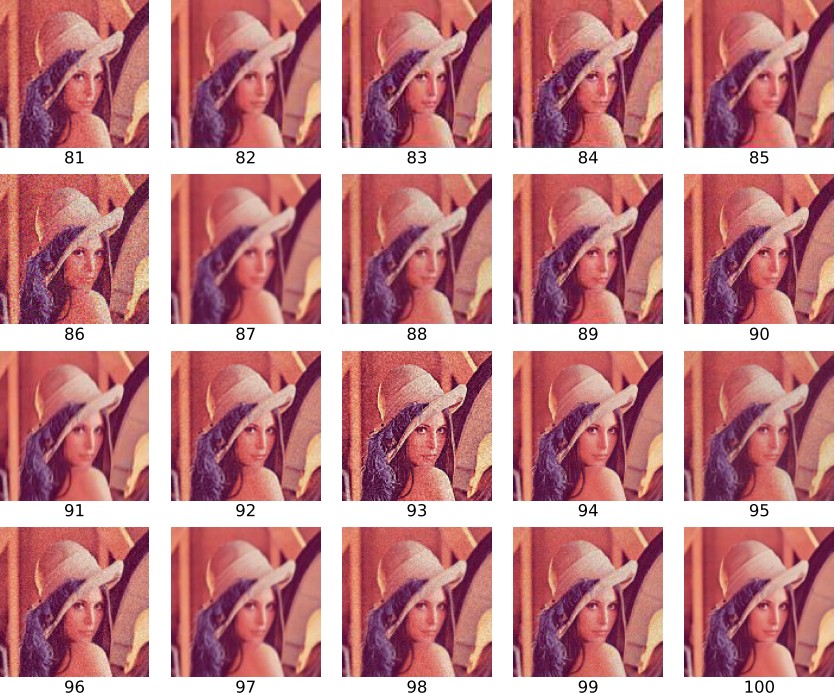

Figure 22: Visualization of the degradation cluster centers [81, 100], sorted by the PSNR of the output of FSRCNN-mz. Best viewed in color.

