# A ADAPTABILITY OF OUR SEAL

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

## C   DETAILS OF CONVENTIONAL EVALUTION

To better understand the biased comparison of conventional evaluation on real-SR, we use common degradation types to illustrate the number of degradation combinations on a simple one-order degradation model. In Tab. 9, the number of degradation combinations is obtained using uniform sampling for each degradation type. It can be seen that the number of degradation combinations can reach up to $1.08 \times 10^5$. Furthermore, RealESRGAN Wang et al. (2021b) adopts a high-order degradation model with complex degradation combinations, such as an-isotropic blur, Poisson noise, gray noise, and *sinc* filter. The combination cases of the high-order degradation models will reach astronomical numbers. As described in the main paper, such a huge space cannot be sufficiently sampled by a few samples due to different combinations of degradation may produce similar visual effects.

Table 9: The number of combination degradation cases on a simple one-order degradation model.

|  | Gaussian Blur | Gaussian noise | Resize | Comression |
|---|---|---|---|---|
| Type | iso | color | nearest, bilinear, bicubic | JPEG |
| Range | Sigma: [0, 2.8] | Sigma: [2, 25] | scale: [0.125, 2] | range: [30, 95] |
| Sampling interval | 0.2 | 2 | 0.1 | 5 |
| Number | 14 | 11 | 54 | 13 |
| Total number | $14 \times 11 \times 54 \times 13 = 1.08 \times 10^5$ | | | |

Inspired by our proposed evaluation framework, which utilizes hundreds of representative test sets to evaluate real-SR models, we create 20 random test sets to further analyze existing evaluation methods. Using a large degradation model, we randomly apply degradations to the images from the DIV2K_val dataset. As shown in Fig. 12, each bar represents the PSNR difference between BSRNet and RealESRNet in a single test set. It reveals that the comparison conclusions are often inconsistent (e.g., -0.22 dB in test set 1 and 0.18 dB in test set 20) or indistinguishable (e.g., -0.01 dB in test set 12) using a single test set. When we average the PSNR results across all test sets, we find an average

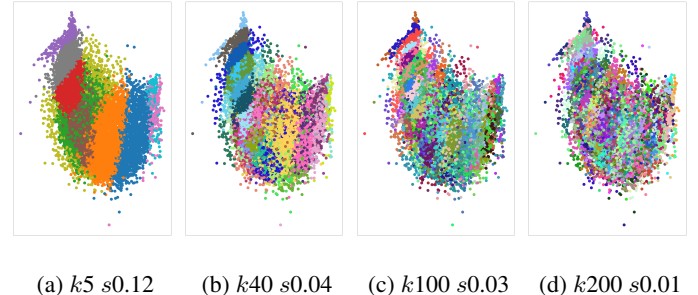

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

### E.2 DEGRADATION CLUSTERING RESULTS

In this section, we present visual results for the *lenna* image, processed with degradation parameters of cluster center [1, 100]. These results are sorted based on the PSNR values of the output from FSRCNN-mz. Fig. 17 showcases the most challenging cases encountered in our study. On the other hand, Fig. 21 highlights the cases that were relatively easier to handle. This comparative analysis provides a clear understanding of the performance range of our proposed evaluation framework.

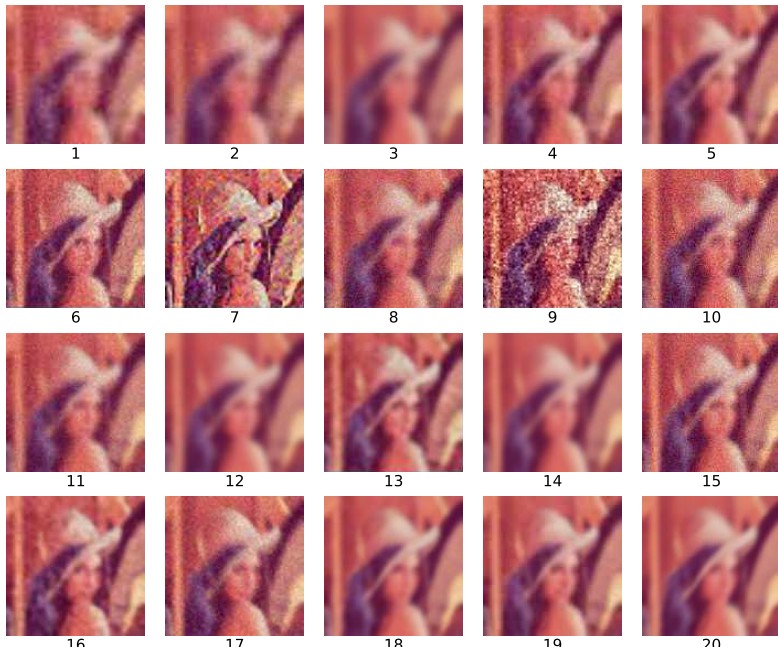

Figure 17: The visual results with the degradation parameters of cluster center [1, 20]. Best viewed in color.

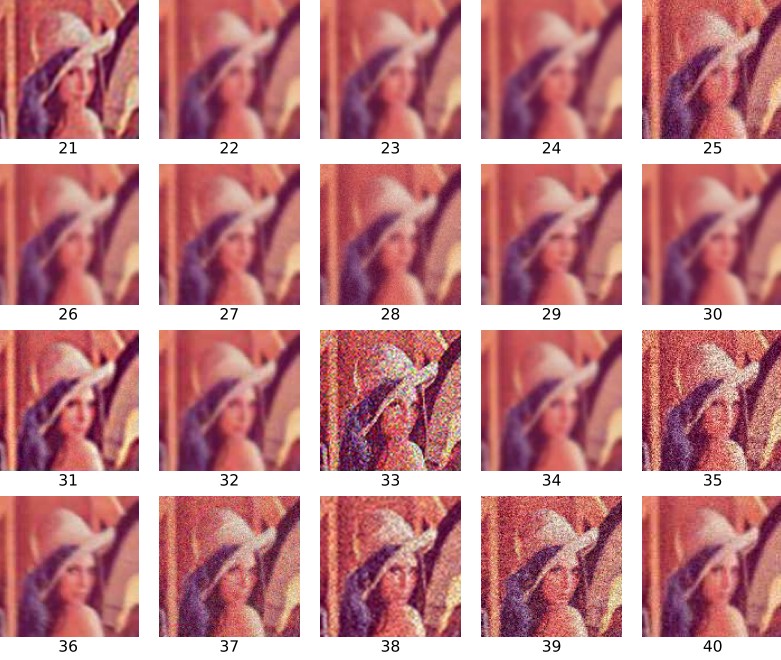

Figure 18: The visual results with the degradation parameters of cluster center [21, 40]. Best viewed in color.

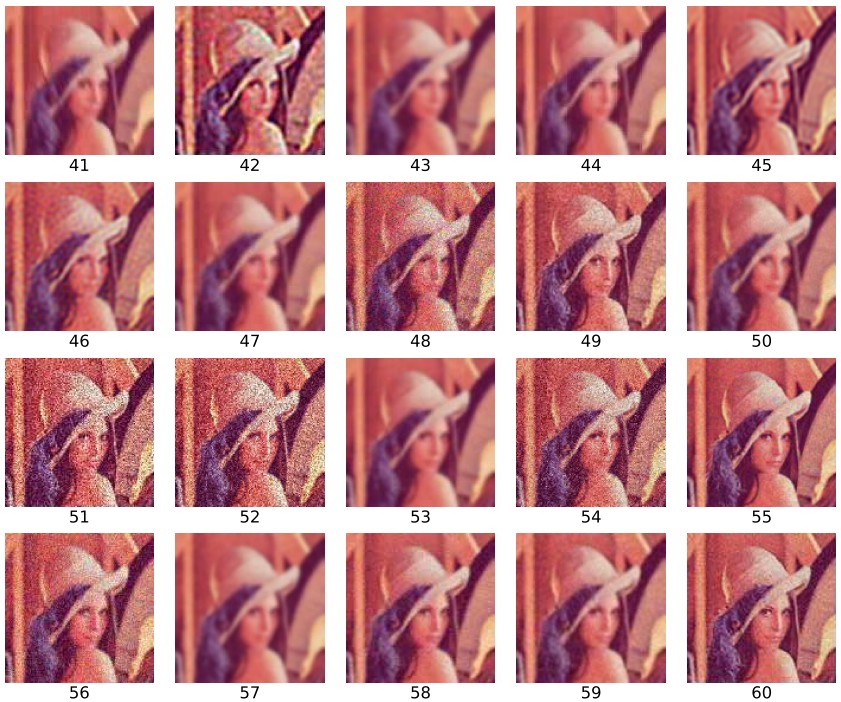

Figure 19: The visual results with the degradation parameters of cluster center [41, 60]. Best viewed in color.

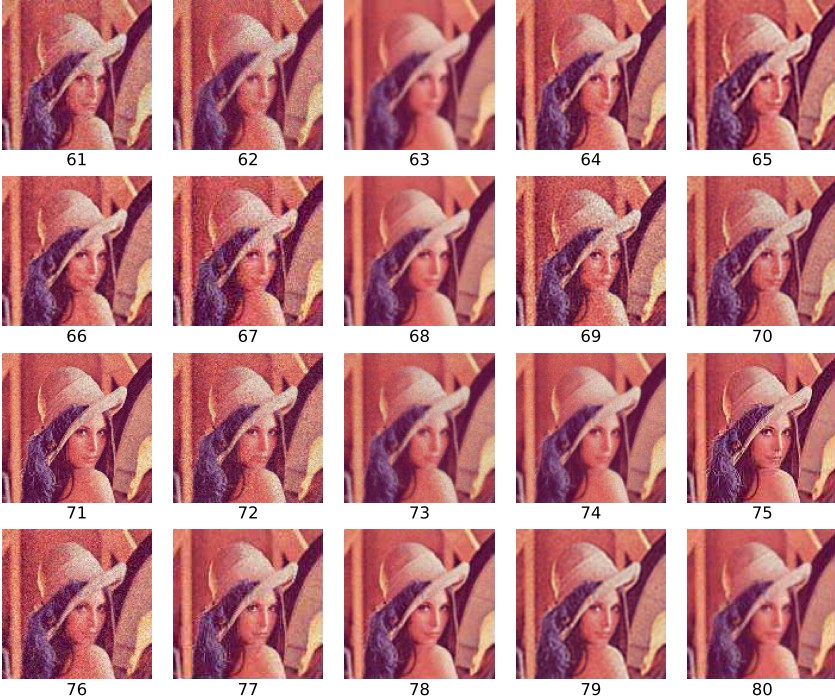

Figure 20: The visual results with the degradation parameters of cluster center [61, 80]. Best viewed in color.

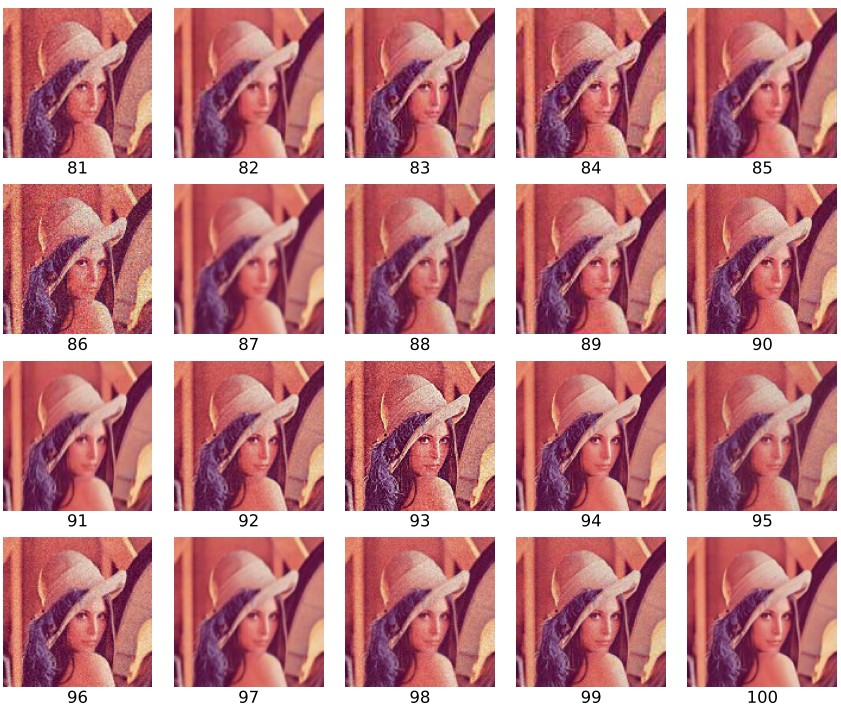

Figure 21: The visual results with the degradation parameters of cluster center [81, 100]. Best viewed in color.