# OpenReview forum: "SEAL: A Framework for Systematic Evaluation of Real-World Super-Resolution"
_ICLR.cc/2024/Conference — ICLR 2024 spotlight_

### Official Review · Reviewer_p1DW · 2023-10-28

**Soundness:** 3 good
**Presentation:** 3 good
**Contribution:** 3 good
**Rating:** 8
**Confidence:** 4

**Summary:**

This work aims to come up with a benchmarking approach to evaluate real SR methods in literature. To this end authors propose to use Acceptance Rate (AR) for coarse grained comparisons and Relative Performance Ratio (RPR) based measures for fine-grained analysis. The motivation and the impact of the proposed metrics are clearly demonstrated with interesting findings on why the previous approaches are misguiding and how the proposed method can alleviate the associated problems.

**Strengths:**

1. The problem tackled in this work has significant practical impact.
2. The proposals are intuitive and supported by experimental evidences.
3. The proposed method comes up with a new and more reliable benchmarking approach for evaluating real-SR methods.

**Weaknesses:**

The reliability of clustering algorithms is measured using purity accuracy.
1. The definition of purity accuracy is fuzzy and will need better explanation.
2. I would advise to demonstrate via visual comparisons that the clusters formed with the proposed method indeed are meaningful to answer the following questions - does a bunch of randomly sampled images from a given cluster look degraded by a comparable amount? Does, random samples from different clusters exhibit contrasting degradation levels?

**Questions:**

Please address the comments under weaknesses

---

> ### Author Response · Authors · 2023-11-20
>
> **1. The definition of purity accuracy is fuzzy and will need better explanation.**
>
> Thank you for bringing this to our attention. We have revised the manuscript to incorporate detailed description of purity accuracy as highlighted in blue on page 17 of the Appendix. Purity [A] is an external criterion (similar to NMI, F measure) that assesses the alignment of the clustering outcome with the actual, known classes. It is defined as:
> $$
> \mbox{purity}(
> \Omega,\mathbb{C}
> ) =
> \frac{1}{N}
> \sum_k \max_j
> \vert\omega_k \cap
> c_j\vert
> $$
> Where $N$ is the total number of data points, $\Omega = \{ \omega_1, \omega_2, \ldots, \omega_K \}$ is the set of clusters and $\mathbb{C} = \{ c_1,c_2,\ldots,c_J \}$ is the set of classes. To compute purity, each cluster is assigned to the class that appears most frequently within that cluster. The accuracy of this assignment is then evaluated by the number of correctly assigned points divided by the total number of points.
>
> [A] Schütze H, Manning C D, Raghavan P. Introduction to information retrieval[M]. Cambridge: Cambridge University Press, 2008.
>
> **2. I would advise to demonstrate via visual comparisons that the clusters formed with the proposed method indeed are meaningful to answer the following questions - does a bunch of randomly sampled images from a given cluster look degraded by a comparable amount? Does, random samples from different clusters exhibit contrasting degradation levels?**
>
> Thank you for the valuable suggestion. In Fig. 17 on page 20 of the Appendix, we’ve demonstrated the visual comparison of samples both between and within clusters. We randomly selected five clusters out of a total of 100 clusters and chose six samples from each cluster. The observation reveals that samples within each cluster display similar degradation patterns, indicating they share a comparable level of restoration difficulty. Conversely, samples belonging to different clusters exhibit contrasting degradation levels. The visual comparison confirms our method is meaningful and effective.

---

### Official Review · Reviewer_JzYv · 2023-10-29

**Soundness:** 3 good
**Presentation:** 3 good
**Contribution:** 3 good
**Rating:** 6
**Confidence:** 2

**Summary:**

In this paper, the authors proposed a new evaluation framework for the evaluation of real-world super-resolution (SR) methods. They first used a clustering-based approach to model a large degradation space, and then designed two new evaluation metrics to assess real-SR models on representative degradation cases. The authors benchmarked existing real-SR methods with the proposed evaluation protocol and presented new observations and insights.

**Strengths:**

The authors handle the issue of fair and comprehensive evaluation of real-SR methods, which can facilitate the development of real-world SR.

**Weaknesses:**

1. In Introduction, the authors argued that random selection may cause significant bias and randomness to the evaluation results. From my point of view, random selection will not cause bias because the posibility of choosing each degradation is identical. The authors are suggested to check this claim to avoid misunderstanding. Same issues exist in Section 5.3.

2. In Section 3.1, there may be a mistake in the calculation of the total combination of degradations. Since the order of degradation types can be switched, the total degradation combination for $s$ degradation type and $k$ degradation levels should be $A^{s}_{s} \times k^{s}$.

3. In Section 3.2, the authors claimed that different combinations of degradation types may have similar visual quality and restoration defficulty. Consequently, the authors designed a clustering-based approach to handle this problem. However, this approach seems intuitive, and experiments should be conducted to validate whether the clustering works as expected, i.e., whether the examples within a cluster have similar visual quality and restoration defficulty whereas the examples in different clusters differ significantly. Only using Fig. 13 for demonstration is not enough.

4. The authors only used one image (i.e., lenna) to generate 100 degradation parameters. It can be observed in Fig. 9 that the image content can affect the final results (0.1-0.3 dB in PSNR). Consequently, have the authors considered using more images to generate more stable and representative clustering centers?

**Questions:**

In Section 5.4, what is the purity accuracy defined as? More details should be explained.

---

> ### Author Response · Authors · 2023-11-20
>
> **1. In Introduction, the authors argued that random selection may cause significant bias and randomness to the evaluation results. From my point of view, random selection will not cause bias because the posibility of choosing each degradation is identical. The authors are suggested to check this claim to avoid misunderstanding. Same issues exist in Section 5.3.**
>
> Thank you for bringing this to our attention. We agree with the reviewer that there is no bias in the selection of degradation cases as they are chosen randomly. When we refer to the results as "biased", we mean to say that the evaluation results are skewed due to the unrepresentative nature of the randomly selected cases. These results only demonstrate the model's performance in these specific cases and do not provide a comprehensive view of its performance across the entire degradation space. To prevent any confusion or misinterpretation, we have modified the phrasing to "inconsistent and potentially misleading" and revised the manuscript accordingly.
>
>
> **2. In Section 3.1, there may be a mistake in the calculation of the total combination of degradations. Since the order of degradation types can be switched, the total degradation combination for $s$ degradation type and $k$ degradation levels should be $A_s^s \times k^s$.**
>
> Thank you for spotting the mistake! Assume there are only $s$ degradation types, and each type contains only $k$ discrete degradation levels. The correct calculation for the total degradation should be $A_s^s \times k^s$. With $s=10$ and $k=10$, the generated degradation space would have a magnitude of $(A_{10}^{10})*10^{10}$, which is $A_{10}^{10}$ times larger than the previous figure that did not consider the order. This number serves as a powerful illustration of the immense scale of the degradation space. We have revised the description on page 3 to reflect this correction.
>
>
> **3. In Section 3.2, the authors claimed that different combinations of degradation types may have similar visual quality and restoration defficulty. Consequently, the authors designed a clustering-based approach to handle this problem. However, this approach seems intuitive, and experiments should be conducted to validate whether the clustering works as expected, i.e., whether the examples within a cluster have similar visual quality and restoration defficulty whereas the examples in different clusters differ significantly. Only using Fig. 13 for demonstration is not enough.**
>
> Thank you for the valuable suggestion. In Fig. 17 on page 20 of the Appendix, we’ve added the visual comparison of samples both between and within clusters. We randomly selected five clusters out of a total of 100 clusters and chose six samples from each cluster. We observed that samples within each cluster exhibit similar degradation patterns, indicating they share a comparable level of restoration difficulty. In contrast, samples belonging to different clusters display remarkably distinct degradation patterns. The results demonstrate that the clustering algorithm worked as expected.
>
>
> **4. The authors only used one image (i.e., lenna) to generate 100 degradation parameters. It can be observed in Fig. 9 that the image content can affect the final results (0.1-0.3 dB in PSNR). Consequently, have the authors considered using more images to generate more stable and representative clustering centers?**
>
> We appreciate your careful reading and insightful comment. Based on your suggestion, we have incorporated a new experiment by utilizing the 5 reference images in Figure 9 for degradation clustering. To accomplish this, we computed the average of the similarity matrices induced by these images. We used the newly identified representative degradation cases to rank GAN-based methods and provided the results in Table 7 on page 13 of the Appendix. Notably, we have observed that this adjustment has led to a more conclusive ranking compared to the results obtained when using a single image (Table 2). In Table 2, we observed that SwinIR and MMRealSR had a very close difference of only 0.01 in the $AR$ metric. Consequently, their ranks needed to be determined using finer metrics such as $RPR_I$. However, in Table 7, we found that SwinIR (AR: 0.86) and MMRealSR (AR: 0.75) could be easily ranked based on the AR metric alone. This suggests that the degradation cases identified by combining the 5 images may indeed be more representative. It also emphasizes the potential benefits of applying enhanced clustering algorithms, which can further enhance the stability and representativeness of our SEAL framework.

---

### Official Review · Reviewer_eBzL · 2023-10-30

**Soundness:** 4 excellent
**Presentation:** 4 excellent
**Contribution:** 4 excellent
**Rating:** 8
**Confidence:** 5

**Summary:**

This paper revisits real-world super-resolution evaluation from a distributional viewpoint, integrating various representative degradation types into the test sets. Previous works only conduct evaluation using average performance on a small set of degradation cases randomly selected from a large space, which often yields biased results. To address this issue, the authors adopt a simple yet effective degradation clustering strategy to select representative degradation. Then, they present an evaluation protocol with two new model-based metrics suitable for real-SR tasks. The experimental results offer a new perspective on real-SR evaluation through the proposed SEAL systematic evaluation framework.

**Strengths:**

1.It is reasonable and interesting to use representative degradation to construct test set for real-SR evaluation.

2.  The authors employ a straightforward yet effective strategy (spectral clustering) for degradation clustering. This approach is not only easy to comprehend, but it also offers potential for further extensions.

3.  The systematic test set offers flexible customization options.

4.  The introduction of new evaluation metrics provides a fresh perspective for real-SR evaluations. These metrics can also complement existing ones, thereby enriching the evaluation strategy.

5.The paper is well-written, and its main idea is easy to follow.

**Weaknesses:**

1.  In Figure 1, the author compares the average performance of two randomly selected test sets for conventional evaluation. If evaluations were conducted on multiple randomized test sets, it could potentially provide a more comprehensive understanding of the model’s performance. It would allow for the observation of performance trends across different test sets and offer insights into the model’s consistency and reliability.

2.  Utilizing a hundred test sets can indeed offer substantial reference points for real-SR evaluation. However, this approach may lead to an increase in the inference time required for evaluation. It would be beneficial for the author to compare the time cost of the existing evaluation method with that of the proposed SEAL evaluation to provide a more comprehensive understanding of their efficiency.

3.  A more detailed explanation of the spectral clustering process, as outlined in Section 3.2, would be beneficial.

**Questions:**

1.  Figure 3 presents a coarse-to-fine evaluation protocol. Initially, the authors utilize AR to validate the effectiveness of the real-SR model in comparison to the acceptance line. Subsequently, they employ coarse metrics to rank the real-SR model. However, actual evaluations may be contingent on user requirements. Could the sequence of fine-grained indicators possibly be adjusted to meet specific needs?

2.  Why is it that the evaluation based on MSE-based real-SR employs multiple smaller models as reference lines, whereas the GAN-based real-SR utilizes a single model as reference lines?

---

> ### Author Response · Authors · 2023-11-20
>
> **1. In Figure 1, the author compares the average performance of two randomly selected test sets for conventional evaluation. If evaluations were conducted on multiple randomized test sets, it could potentially provide a more comprehensive understanding of the model’s performance. It would allow for the observation of performance trends across different test sets and offer insights into the model’s consistency and reliability.**
>
>
> We appreciate your valuable suggestion. We already provided the results on 20 randomized test sets in Figure 12 on page 16 of the Appendix. These results demonstrate a high level of randomness and significant variance. Across these 20 test sets, BSRNet and RealESRNet each outperform the other in approximately half of the cases, with their performance gap ranging from -0.22 to 0.18. It is difficult to definitively determine which method is superior. This observation further highlights the inadequacy of using randomized test sets for evaluation.
>
> **2. Utilizing a hundred test sets can indeed offer substantial reference points for real-SR evaluation. However, this approach may lead to an increase in the inference time required for evaluation. It would be beneficial for the author to compare the time cost of the existing evaluation method with that of the proposed SEAL evaluation to provide a more comprehensive understanding of their efficiency.**
>
> Thank you for the thoughtful suggestion. We have included a comparison of the time cost between the conventional evaluation method and our approach in Table 9 on page 15 of the Appendix. As expected, our approach does result in an increase in inference time, which scales linearly with the number of identified representative degradation cases (e.g., 100 in our experiments). However, since the inference time remains within acceptable limits, we believe this is a worthwhile tradeoff between evaluation efficiency and quality.
>
> **Comparison of the time cost between the conventional evaluation method and our approach using RRDBNet.**
>
> |                   | inference time [s] | PSNR run-time [s] | $AR, RPR$ run-time [s] |
> |-------------------|-------------------:|------------------:|-------------------------:|
> | Set14             | 4.22               | 0.52              | 0.013                    |
> | Set14-SE (ours)   | 382.74             | 49.47             | 0.013                    |
>
> **3. A more detailed explanation of the spectral clustering process, as outlined in Section 3.2, would be beneficial.**
>
> Thank you for the thoughtful question. Due to space limitations, we provided a detailed description of degradation clustering on page 16 of the Appendix.
>
> **4. Figure 3 presents a coarse-to-fine evaluation protocol. Initially, the authors utilize AR to validate the effectiveness of the real-SR model in comparison to the acceptance line. Subsequently, they employ coarse metrics to rank the real-SR model. However, actual evaluations may be contingent on user requirements. Could the sequence of fine-grained indicators possibly be adjusted to meet specific needs?**
>
> We appreciate your insightful question. Similar to other components of our SEAL framework, the proposed coarse-to-fine evaluation metric can be adjusted according to the requirements and preferences of users. For instance, if enhancing the performance of acceptable degradation tasks is a priority, the user can choose to place $RPR_A$ before $RPR_I$.
>
> **5. Why is it that the evaluation based on MSE-based real-SR employs multiple smaller models as reference lines, whereas the GAN-based real-SR utilizes a single model as reference lines?**
>
> Thank you for the careful reading and thoughtful question. Our SEAL framework provides flexibility in aligning with specific requirements. Users can make different design choices when selecting the appropriate reference lines that best suit their needs. In the GAN-based setting, generating high-quality texture details with a small network can be challenging. To address this, we utilize RealESRGAN and RealHATGAN as reference models for acceptance and excellence, respectively. However, training these models individually on 100 tasks would be time-consuming due to their large size. As a result, we use their performance on the 100 tasks as reference points without further fine-tuning them on these tasks.

---

### Official Review · Reviewer_xKgq · 2023-11-01

**Soundness:** 3 good
**Presentation:** 3 good
**Contribution:** 3 good
**Rating:** 6
**Confidence:** 4

**Summary:**

The paper presents a framework for evaluating blind single-image super-resolution techniques against multiple different degradation types.  The process generates a large random sampling of degradations and then clusters the resulting images based on histograms. Each cluster represents functionally similar degradations, regardless of the process of obtaining them. The authors can then generate a diverse and comprehensive test set by sampling from the degradations represented by each cluster.

The authors also use a fully automated method of evaluating new networks relative to two existing networks that play the role of baseline (acceptable) and best-in-class (excellent). By evaluating each test image relative to these two networks, they can provide more meaningful statistics about the robustness of each network to the range of potential degradations.

**Strengths:**

The paper addresses an issue in blind single-image super-resolution that has not yet been fully solved: how to effectively evaluate performance. It is clear from prior work that PSNR is insufficient, since it does not always agree with human perception.  SSIM and perceptual distance provide potentially more informative metrics, but using aggregate averages does not convey strength across different types of degradation.

The use of other networks to provide baseline (acceptable) and state-of-the-art (excellent) performance is an interesting approach. One downside is the continued reliance on PSNR, which has a hard time capturing fine textures or visual structures.

The clustering-based approach to identifying diverse degradations is an interesting idea.

**Weaknesses:**

The authors use histograms as the basis for clustering degradations. However, the form of the histograms does not appear to be given in the paper (sections 3.2, 3.3).  Are these histograms of r, g, and b separately?  Is this a 2-D chromaticity histogram, a 3-D RGB histogram, or a set of histograms of filter bank outputs?  It's not clear that all of these would be equally effective at capturing diverse degradations.

The sole use of PSNR as a way to rank-order results may not be the best choice.  PSNR rankings don't always reflect human rankings.

**Questions:**

Why use PSNR as the basis for whether an image is Acceptable?  Why not SSIM, perceptual distance, or some combination of the three?  Do you get different rankings with different metrics under this evaluation system?

The histograms used in the clustering are histograms of what?

---

> ### Author Response · Authors · 2023-11-20
>
> **1. The histograms used in the clustering are histograms of what? The authors use histograms as the basis for clustering degradations. However, the form of the histograms does not appear to be given in the paper (sections 3.2, 3.3). Are these histograms of r, g, and b separately? Is this a 2-D chromaticity histogram, a 3-D RGB histogram, or a set of histograms of filter bank outputs? It's not clear that all of these would be equally effective at capturing diverse degradations.**
>
> Thank you for the question. The histograms of R, G, and B are generated separately and then concatenated into a single vector to represent the degraded image. We have included this information on page 16 of the new version and highlighted it in blue.
>
> **2. The sole use of PSNR as a way to rank-order results may not be the best choice. PSNR rankings don't always reflect human rankings. Why use PSNR as the basis for whether an image is Acceptable? Why not SSIM, perceptual distance, or some combination of the three? Do you get different rankings with different metrics under this evaluation system?**
>
> Thank you for your insightful question. Our SEAL framework offers the flexibility to incorporate various IQA metrics, allowing users to customize the evaluation according to their specific requirements and objectives. In our experiments, we have explored and utilized multiple IQA metrics. Specifically, for MSE-based real-SR methods, we utilize the widely used PSNR metric. For GAN-based real-SR methods, we employ the perceptual metric LPIPS, as demonstrated in Table 2 on page 7. Additionally, we present evaluation results using the SSIM metric in Table 6 on page 13 of the Appendix.
>
> Indeed, different metrics may yield different rankings. For instance, BSRNet ranks first in PSNR, while RealESRNet holds the first position in SSIM. This is because RealESRNet is trained with sharpened ground truth, which favors structural information. This variation in rankings highlights the utilization of different metrics in our evaluation framework can effectively capture and reflect the unique characteristics of real-SR methods.

---

### Author Response · Authors · 2023-11-20
**General Response to All Reviewers**

We sincerely thank all the reviewers for their careful reading and insightful, constructive feedback, which has significantly contributed to enhancing the quality of our work. We have made revisions to the manuscript in line with the suggestions and have included additional experiments. For convenience, we have merged the Appendix and the main paper into a single PDF file.

We have responded individually to each reviewer's comments, and we are more than willing to address any additional questions or concerns that may arise.

---

### Meta-Review · Area_Chair_FoNR · 2023-12-10

**Metareview:**

The authors tackle the problem of benchmarking real super-resolution methods and propose a framework for that. Representative degradation cases are obtained through clustering and the evaluation protocol includes two new measures: acceptance rate (AR) and relative performance ratio (RPR). An analysis is conducted on the existing methods and a new strong baseline for Real SR is proposed.
All the reviewers are generally positive about the paper and its contributions. The authors provided satisfactory responses to the reviewers and can be entrusted to further refine their work.
There is a strong need for reliable benchmarks for real SR and this work is a good step in the right direction that invites to further discussion, research and developments.

**Justification For Why Not Higher Score:**

The topic addressed and the contributions in this paper are of interest for super-resolution / restoration community, but not necessarily relevant to the wider ICLR community. It is more of an applied, benchmarking paper without a significant machine learning or fundamental/ theoretical contribution.

**Justification For Why Not Lower Score:**

The paper makes significant contributions towards the benchmarking of real SR methods. That has a strong impact potential and is of special interest for the super-resolution / restoration community. In comparison with a poster, a spotlight draws more attention and invites to further discussion, research and developments.

---

### Decision · Program_Chairs · 2024-01-16

Accept (spotlight)